

# Bohdi: Heterogeneous LLM Fusion with Automatic Data Exploration

**Junqi Gao**[1],[*] **Zhichang Guo**[1], **Dazhi Zhang**[1], **Dong Li**[1]
**Runze Liu**[3], **Pengfei Li**[1,5], **Kai Tian**[4], **Biqing Qi**[2],[†]

[1] School of Mathematics, Harbin Institute of Technology
[2] Shanghai Artificial Intelligence Laboratory
[3] Tsinghua Shenzhen International Graduate School, Tsinghua University
[4] Department of Electronic Engineering, Tsinghua University
[5] Shanghai Innovation Institute
gjunqi97@gmail.com,qibiqing7@gmail.com

## Abstract

Heterogeneous Large Language Model (LLM) fusion integrates the strengths of multiple source LLMs with different architectures into a target LLM with low computational overhead. While promising, existing methods suffer from two major limitations: 1) **reliance on real data from limited domain** for knowledge fusion, preventing the target LLM from fully acquiring knowledge across diverse domains, and 2) **fixed data allocation proportions** across domains, failing to dynamically adjust according to the target LLM's varying capabilities across domains, leading to a capability imbalance. To overcome these limitations, we propose Bohdi, a synthetic-data-only heterogeneous LLM fusion framework. Through the organization of knowledge domains into a hierarchical tree structure, Bohdi enables automatic domain exploration and multi-domain data generation through multi-model collaboration, thereby comprehensively extracting knowledge from source LLMs. By formalizing domain expansion and data sampling proportion allocation on the knowledge tree as a Hierarchical Multi-Armed Bandit problem, Bohdi leverages the designed DynaBranches mechanism to adaptively adjust sampling proportions based on the target LLM's performance feedback across domains. Integrated with our proposed Introspection-Rebirth (IR) mechanism, DynaBranches dynamically tracks capability shifts during target LLM's updates via Sliding Window Binomial Likelihood Ratio Testing (SWBLRT), further enhancing its online adaptation capability. Comparative experimental results on a comprehensive suite of benchmarks demonstrate that Bohdi significantly outperforms existing baselines on multiple target LLMs, exhibits higher data efficiency, and virtually eliminates the imbalance in the target LLM's capabilities. Our code is available at ⏻ Bohdi.

## 1 Introduction

With the rapid advancement of Large Language Model (LLM) [1, 2] capabilities and the thriving development of open-source communities, both enterprises and individual users are actively engaged in building their own LLMs. As an efficient solution, LLM fusion [3, 4, 5, 6] technology consolidates the advantages of multiple source LLMs into a more compact target LLM without the need for

---

[*]This work was done during his internship at Shanghai Artificial Intelligence Laboratory.
[†]Corresponding author.

39th Conference on Neural Information Processing Systems (NeurIPS 2025).

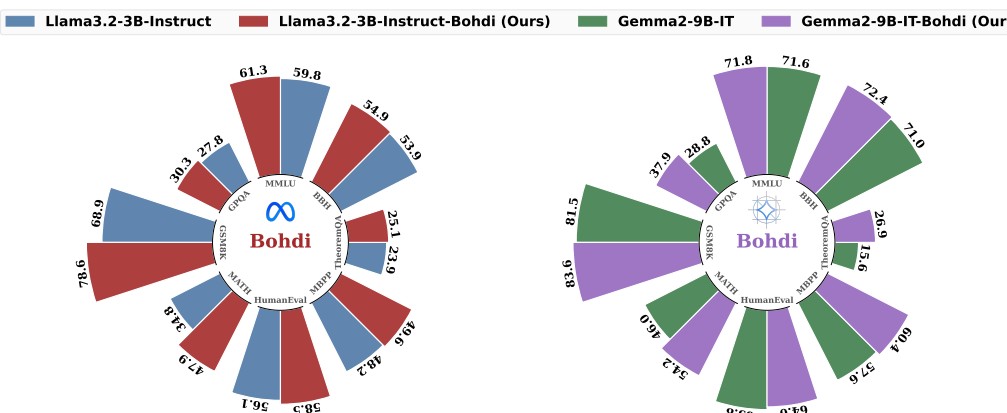

Figure 1: Results of Bohdi across various benchmarks, using Llama-3.2-3B-Instruct (left) and Gemma2-9B-IT (right) as target LLMs.

pretraining from scratch or post-training based on large amounts of data. This significantly shortens the development cycle of private LLMs and reduces the required computing resources.

Due to the differences in architecture and size among open-source LLMs, there is always a heterogeneity between source and target LLMs. This limits knowledge fusion through direct parameter merging approaches [3, 4] and has driven research into heterogeneous LLM fusion, with methods mainly divided into two types: Explicit Fusion (EF) and Implicit Fusion (IF). EF methods [5, 6] involve an initial vocabulary alignment, followed by supervised training of the target LLM using output probability distributions collected from the source LLMs on a given dataset. However, the vocabulary alignment process introduces errors and noise [7], compromising the target LLM's learning stability. In contrast, IF methods [7, 8, 9] collect responses from the source LLMs and uses a external reward LLM to select high-quality response for end-to-end training of the target LLM, thereby bypassing the vocabulary alignment process and enabling more stable knowledge injection.

While demonstrating promising performance, current heterogeneous LLM fusion methods still face two key limitations: 1) **Limited Domain Coverage**: Due to the scarcity of real data in systematically finely-grained domains, current approaches are limited to relying on open-source data from a few fixed domains (e.g., mathematics, coding). This prevents the target LLM from fully acquiring knowledge from source LLMs across diverse domains, thereby restricting comprehensive performance improvement. 2) **Rigid Data Allocation**: Current methods pre-specify the data proportions allocated to each domain. According to the "Buckets effect", an ideal data allocation should emphasize domains where the target LLM performs poorly. However, fixed data proportion assignments cannot adapt to the varying data proportion needs of different target LLMs, leading to a capability imbalance where improvements in some capabilities come at the expense of others.

Addressing the aforementioned limitations, we propose BOHDI, a heterogeneous LLM fusion framework. By constructing a hierarchical knowledge tree, Bohdi systematically manages multi-domain knowledge. Equipped with the designed Sprout and Harvest operations, Bohdi enables dynamic exploration of new domains through multi-model collaboration and automatic generation of data from each domain, thus achieving more comprehensive domain expansion without relying on real data. To achieve adaptive adjustment of domain data proportions, we formalize dynamic domain expansion and sampling proportion allocation on the knowledge tree as a Hierarchical Multi-Armed Bandit (HMAB) problem. We then propose DynaBranches, which employs Thompson Sampling (TS) [10] to efficiently sample multi-domain data and adaptively update sampling proportions for each domain through the Sprout and Harvest operations. Combined with our proposed Introspection-Rebirth (IR) mechanism, DynaBranches uses the designed Sliding Window Binomial Likelihood Ratio Test (SWBLRT) to dynamically detect changes in domain capabilities during the target LLM's updates, thereby avoiding over-reliance on outdated observations in the dynamic sampling process.

Integrating the aforementioned components and mechanisms, Bohdi's fusion process is modeled as an iterative two-phase optimization: **MEDITATION** and **ENLIGHTENMENT**. The Meditation phase focuses on domain exploration, data collection, and adjustment of domain data proportions, while the Enlightenment phase trains the target LLM based on the current optimal sampling proportions.

Through the alternating iterative execution of these two phases, Bohdi achieves dynamic knowledge domain expansion and multi-domain source LLM knowledge acquisition without relying on any real data. It also enables online adaptive adjustment of domain data proportions, effectively injecting multi-domain knowledge from multiple source LLMs. Comprehensive comparative experiments across a range of benchmarks demonstrate that Bohdi achieves significant performance improvements over existing baselines on multiple target LLMs without relying on real data, while also exhibiting higher data efficiency and virtually eliminates the imbalance in the target LLM's capabilities. The contributions of our work can be summarized as follows:

- We propose Bohdi, the first synthetic-data-only heterogeneous LLM fusion framework that manages multi-domain knowledge through a hierarchical knowledge tree, while enabling automated domain expansion and data collection via multi-model collaboration.

- By formalizing the problem of domain expansion and sampling proportion allocation as a HMAB problem, we propose DynaBranches with integrated IR mechanism for dynamic capability monitoring and efficient online multi-domain proportion adaptation.

- Through the constructed Meditation-Enlightenment iteration, Bohdi automatically performs multi-domain source LLM knowledge acquisition while adaptively adjusting cross-domain data proportions online, efficiently injecting source LLM knowledge and effectively preventing the imbalance in target LLM's capabilities.

## 2 Methodology

### 2.1 Overall Objective of The Bohdi Framework

We first define the overall framework of Bohdi. Let the set of source LLMs be denoted as $\mathcal{S} = \{\mathcal{M}_k^S\}_{k=1}^K$, each $\mathcal{M}_k^S(\cdot, \boldsymbol{\theta}_k^S) : \mathcal{Q} \to \mathcal{A}$ ($k$ denotes the subscript index) is an LLM parameterized by $\boldsymbol{\theta}_k^S$, which accepts a question $q$ from the set of queries $\mathcal{Q}$ and returns an answer $a \in \mathcal{A}$, where $\mathcal{A}$ is the set of answers. To fully integrate the strengths of the $K$ source LLMs into the target LLM $\mathcal{M}^T$ parameterized by $\boldsymbol{\theta}$, we aim to ensure that for any domain $\mathcal{D}$, the target LLM performs no worse than the best source LLM in $\mathcal{S}$ for that domain. This objective can be formally expressed as:

$$\boldsymbol{\theta} = \arg\min_{\boldsymbol{\theta}} \mathbb{E}_{\mathcal{D}} \mathbb{E}_{q \in \mathcal{Q}^{\mathcal{D}}} \mathbb{1}\left( V\left(q, \mathcal{M}^T\left(q, \boldsymbol{\theta}\right)\right) < \max_{1 \le k \le K} V\left(q, \mathcal{M}_i^S\left(q, \boldsymbol{\theta}_k^S\right)\right) \right), \quad (1)$$

where $V(q, a)$ measures the quality of the answer $a$ in response to the question $q$, $\mathcal{Q}^{\mathcal{D}}$ denotes the question space belonging to domain $\mathcal{D}$. Given the broad range of problem domains, it is not feasible to list them all at once. Moreover, obtaining real data for each specialized domains is challenging. Additionally, directly assigning static fixed proportions to data from each domain can easily lead to a severe capability imbalance. Therefore, we aim for the domains in Problem (1) to dynamically expand and automatically generate corresponding domain data, while also enabling online adjustment of data proportion across different domains. To this end, we transform the problem in Eq. (1) into the following two-phase optimization problem:

$$\textcolor{red}{\mathcal{D}^{(t+1)}} = \arg\max_{\textcolor{red}{\mathcal{D}^{(t+1)}}} \mathbb{E}_{q \in \mathcal{Q}^{\mathcal{D}^{(t+1)}}} \mathbb{1}\left( V\left(q, \mathcal{M}^T\left(q, \boldsymbol{\theta}^{(t)}\right)\right) < \max_{1 \le k \le K} V\left(q, \mathcal{M}_k^S\left(q, \boldsymbol{\theta}_i^S\right)\right) \right), \quad (2)$$

$$\textcolor{red}{\boldsymbol{\theta}^{(t+1)}} = \arg\min_{\textcolor{red}{\boldsymbol{\theta}^{(t+1)}}} \mathbb{E}_{q \in \mathcal{Q}^{\mathcal{D}^{(t+1)}}} \mathbb{1}\left( V\left(q, \mathcal{M}^T\left(q, \boldsymbol{\theta}^{(t+1)}\right)\right) < \max_{1 \le k \le K} V\left(q, \mathcal{M}_k^S\left(q, \boldsymbol{\theta}_i^S\right)\right) \right), \quad (3)$$

where $\mathcal{D}^{(t)}$ and $\boldsymbol{\theta}^{(t)}$ denote the problem domain and target LLM parameters in the $t$-th round of dynamic iteration, respectively. The fusion process of Bohdi consists of the **Meditation** phase formalized by (2) and the **Enlightenment** phase formalized by (3), carried out in sequence. The Meditation phase performs dynamic domain expansion and online adjustment of sampling proportion for each domain, while collecting question-answer data for the corresponding domain. The Enlightenment process trains $\mathcal{M}^T$ based on the collected multi-domain data to inject knowledge from multiple source models into the target LLM.

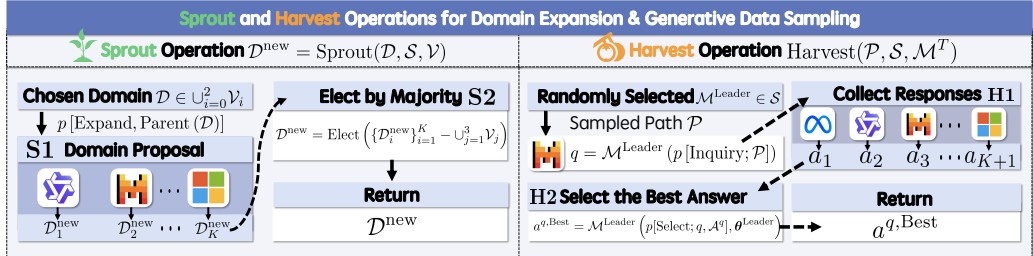

Figure 2: Flowchart of the proposed Sprout (left) and Harvest (right) operations.

## 2.2 Dynamic Domain Exploration

To enable dynamic knowledge domain exploration, a structured organization of knowledge that permits further exploration and data sampling is necessary. Therefore, we introduce the following components:

**Structural Knowledge Tree** $\mathcal{T}$   To provide a more structured organization of problem domains, we employ a tree-like structure in the format of [*Main→Secondary→Sub*]. All question domains originate from the root domain [*Root*], and are constructed in the form $[\mathcal{D}_{1,i} \to \mathcal{D}_{2,j} \to \mathcal{D}_{3,k}]$ $(1 \le i \le N_1, 1 \le j \le N_2, 1 \le k \le N_3)$, where $N_1$, $N_2$, and $N_3$ denote the number of optional domains at levels 1, 2, and 3 respectively. Then we define a knowledge tree $\mathcal{T}(\mathcal{V}, \mathcal{E})$, where

- $\mathcal{V} = \{\mathcal{V}_0, \mathcal{V}_1, \mathcal{V}_2, \mathcal{V}_3\}$ is a hierarchical domain set satisfying $\forall 1 \le i \le 3$, $\mathcal{V}_i = \{\mathcal{D}_{i,j}\}_{j=1}^{N_i} \cup \{\mathcal{D}_{i,\text{unk}}\}$, with $\mathcal{D}_{i,\text{unk}}$ representing the unexplored unknown (denoted as "unk") domain, and $\mathcal{V}_0 = \{\mathcal{D}_{0,0}\}$ as the sole root domain.
- $\mathcal{E} = \{(\mathcal{D}_{i,j}, \mathcal{D}_{i+1,k}) \mid \mathcal{D}_{i,j} \in \mathcal{V}_i, \mathcal{D}_{i+1,k} \in \mathcal{V}_{i+1}$ is a child domain of $\mathcal{D}_{i,j}, i \in \{0,1,2\}\}$ represents the set of edges.

**Sprout Operation for Domain Expansion** On the defined knowledge tree, we can sample a path from the root to a level-3 domain to determine the question domain. For domain $\mathcal{D}_{i,j}$ satisfying $0 \le i \le 2$, we introduce the Sprout operation, as illustrated on the left in Fig. 2, to propose a new domain $\mathcal{D}^{\text{new}} = \text{Sprout}(\mathcal{D}, \mathcal{S}, \mathcal{V})$, with the steps as follows (for detailed descriptions of these two operations, please refer to Appendix B.1):

**S1** Collect the proposed domains $\mathcal{D}_1^{\text{new}}, \mathcal{D}_2^{\text{new}}, \ldots, \mathcal{D}_K^{\text{new}}$ from each source model $\mathcal{M} \in \mathcal{S}$.

**S2** Remove the proposed domains $\mathcal{D}_i^{\text{new}}$ that belong to the existing domain set $\cup_{j=1}^3 \mathcal{V}_j$, and elect the most frequently proposed domain $\mathcal{D}^{\text{new}}$ from the remaining candidates to be added to $\mathcal{D}_i$.

**Harvest Operation for Generative Data Sampling** For any path $\mathcal{P} = [\mathcal{D}_{0,0} \to \mathcal{D}_{1,i} \to \mathcal{D}_{2,j} \to \mathcal{D}_{3,k}]$ sampled on $\mathcal{T}$, we need to obtain question-answer pairs $(q, a)$ belonging to the sub-domain $\mathcal{D}_{3,k}$ on this path to simulate the question space $\mathcal{Q}^{\mathcal{D}_{3,k}}$ and provide corresponding evaluations to assess the quality of each LLM's response $V(q, a)$ to the question $q$. However, due to the highly granular professional division of paths and sub-domains in the knowledge tree, existing open-source datasets generally lack a matching fine-grained annotation system, making it difficult to directly support sampling for these specific domains. To this end, we introduce the Harvest operation, as shown on the right in Fig. 2, to generate the question-answer pair $(q, a) = \text{Harvest}(\mathcal{P}, \mathcal{S}, \mathcal{M}^T)$ and add it to the question-answer pair set $\Omega^{\mathcal{D}_{3,k}}$ corresponding to domain $\mathcal{D}_{3,k}$ for training purposes:

**H1** Collect the responses $a_1, a_2, \ldots, a_{K+1}$ from all $K$ source models and the target model.

**H2** Randomly select a Leader model $\mathcal{M}^{\text{Leader}} \in \mathcal{S}$ from the set of source models to choose the best answer $a^{q,\text{Best}}$ among all the responses $\{a_i\}_{i=1}^{K+1}$.

With clearly defined problem domains and empirical observations of each LLM's performance across these domains, the problem in the Meditation phase now can transforms into a decision-making process for selecting problem domains based on the observed performance of both source and target LLMs as feedback. Consequently, we need to integrate these empirical observations into our decision-making considerations. In the subsequent discussion, we will demonstrate that, given the component designs established earlier, the domain selection process in the Meditation phase can be naturally reformulated as a HMAB problem.

Firstly, based on the defined answer quality score $V(q, a)$, the objective in the Meditation phase Eq. (2) can be transformed into

$$\mathcal{D}^{(t+1)} = \arg\max_{\mathcal{D}^{(t+1)}} \mathbb{E}_{q \in \mathcal{Q}^{\mathcal{D}^{(t+1)}}} r\left(q, \mathcal{M}^T, \{\mathcal{M}_k^S\}_{k=1}^K\right), \tag{4}$$

where

$$r\left(q, \mathcal{M}^T, \{\mathcal{M}_k^S\}_{k=1}^K\right) = \begin{cases} 1, & \text{if } \exists k \in \{1, 2, \ldots, K\}, \text{s.t. } a^{q,\text{Best}} = \mathcal{M}_k^S\left(q, \boldsymbol{\theta}_k^S\right) \\ 0, & \text{if } a^{q,\text{Best}} = \mathcal{M}^T\left(q, \boldsymbol{\theta}\right) \end{cases} \tag{5}$$

serves as the reward function that evaluates the quality of the sampled question $q$, denoted as $r^{\mathcal{D}^{(t+1)}}$. When $r^{\mathcal{D}^{(t+1)}} = 1$, it indicates that under the current sampling, the target LLM's performance in domain $\mathcal{D}^{(t+1)}$ is inferior to that of the source LLMs, meaning the corresponding question-answer pair $(q, a^{q,\text{Best}})$ is worth learning for the target LLM. With the objective above, the path selection process on knowledge tree $\mathcal{T}$ can be formulated as a HMAB problem:

$$\begin{cases} \text{Global-Level: HMAB}\left(\mathcal{T}\right) = \left\langle \{\text{MAB}\left(\mathcal{D}\right)\}_{\mathcal{D} \in \cup_{i=0}^2 \mathcal{V}_i}, \{\mathcal{R}\left(r \mid \mathcal{D}', \lambda_{\mathcal{D}'}\right)\}_{\mathcal{D}' \in \cup_{i=1}^3 \mathcal{V}_i}, \mathcal{T} \right\rangle, \\ \text{Local-Level: MAB}\left(\mathcal{D}\right) = \left\langle \mathcal{C}\left(\mathcal{D}, \mathcal{T}\right), \{\mathcal{R}\left(r \mid \mathcal{D}', \lambda_{\mathcal{D}'}\right)\}_{\mathcal{D}' \in \mathcal{C}(\mathcal{D}, \mathcal{T})} \right\rangle, \forall \mathcal{D} \in \cup_{i=1}^3 \mathcal{V}_i, \end{cases} \tag{6}$$

that is, for any domain $\mathcal{D}_{i,j}$ on $\mathcal{T}$, we define a MAB instance where the arm set is $\mathcal{C}(\mathcal{D}_{i,j}, \mathcal{T}) = \{\mathcal{D}_{i+1,k} \mid \mathcal{D}_{i+1,k} \in \mathcal{V}_{i+1} \text{ and } (\mathcal{D}_{i,j}, \mathcal{D}_{i+1,k}) \in \mathcal{E}\}$, representing the set of child domains under $\mathcal{D}_{i,j}$. $\mathcal{R}(r \mid \mathcal{D}', \lambda_{\mathcal{D}'})$ denotes the reward distribution for selecting domain $\mathcal{D}'$. With the reward defined in Eq. (5), it is a Bernoulli distribution with parameter $\lambda_{\mathcal{D}'}$, i.e., $\mathbb{P}(r = 1 \mid \mathcal{D}') = \lambda_{\mathcal{D}'}$. Thus Eq. (4) can be further simplified to $\mathcal{D}^{(t+1)} = \arg\max_{\mathcal{D}^{(t+1)}} \lambda_{\mathcal{D}^{(t+1)}}$.

Each data sampling process in HMAB($\mathcal{T}$) begins at the root domain $\mathcal{D}_{0,0}$ and sequentially selects arms through three MAB instances: MAB($\mathcal{D}_{0,0}$), MAB($\mathcal{D}_{1,i}$), and MAB($\mathcal{D}_{2,j}$), thereby sampling a path $\mathcal{P} = [\mathcal{D}_{0,0} \to \mathcal{D}_{1,i} \to \mathcal{D}_{2,j} \to \mathcal{D}_{3,k}]$, then the question-answer pairs are sampled for the target LLM training, while obtaining the rewards for updating the parameters of each MAB along the path. Thus the problem of domain exploration and selection in the Meditation phase is now formalized as a sampling problem on HMAB($\mathcal{T}$). By adjusting the reward distribution parameters on each arm (we use the term "arm" to refer to the corresponding optional child domains in the following sections), we can achieve the adjustment of sampling proportions across different domains.

## 2.3 Online Adaptive Sampling

Next, we introduce the DynaBranches and IR mechanisms in sequence to achieve efficient sampling and online adaptive sampling proportion updates on HMAB $(\mathcal{T})$.

**DynaBranches for Efficient Adaptive Sampling** To ensure the sampling efficiency and the adaptiveness of sampling proportions, batch sampling must be conducted while ensuring sampling diversity, and reward distribution must be updated based on batched reward feedback. To this end, we propose DynaBranches, which specifies prior distributions for the reward distribution parameters of each domain. Leveraging the TS algorithm [10], it samples paths on the HMAB and adjusts the posterior based on feedback to achieve adaptive tuning of the sampling proportions. This process models the reward distribution parameters as random variables to introduce randomness into the sampling process, thereby achieving parallel sampling while ensuring the diversity of sampled domains.

Specifically, for an arm $\mathcal{D}$ that has been sampled $n$ times, the cumulative reward distribution observations $\sum_{i=1}^n r_i^{\mathcal{D}}$ follow a binomial distribution, where $r_i^{\mathcal{D}}$ is the observed reward for the $i$-th sampling in that domain. This makes the Beta distribution a conjugate prior for $\lambda_{\mathcal{D}}$ [11]. Therefore, we assign the prior distribution of the reward distribution parameter $\lambda_{\mathcal{D}}$ for each domain $\mathcal{D}$ as Beta(1, 1), which is equivalent to a uniform distribution on $[0, 1]$. Consequently, the posterior of $\lambda_{\mathcal{D}}$ remains in the form of a Beta distribution Beta($\alpha_{\mathcal{D}}, \beta_{\mathcal{D}}$), where $\alpha_{\mathcal{D}} = 1 + \sum_{i=1}^n r_i^{\mathcal{D}}$ and $\beta_{\mathcal{D}} = 1 + n - \sum_{i=1}^n r_i^{\mathcal{D}}$. Then the execution process of DynaBranches can be summarized in the following two steps:

*Step1: Batchwise Path Sampling.* In this step, we sample $B$ paths $\{\mathcal{P}^b\}_{b=1}^B$ on HMAB($\mathcal{T}$). For each path $\mathcal{P}^b$, we initialize it from the root domain as $\mathcal{P}^b = [\mathcal{D}_0^b]$, where $\mathcal{D}_0^b = \mathcal{D}_{0,0}$, and then sample its child domains for each $\mathcal{D}_i^b$ ($0 \le i \le 2$) according to the following two sub-steps until obtaining the complete path $\mathcal{P}^b = [\mathcal{D}_0^b, \mathcal{D}_1^b, \mathcal{D}_2^b, \mathcal{D}_3^b]$:

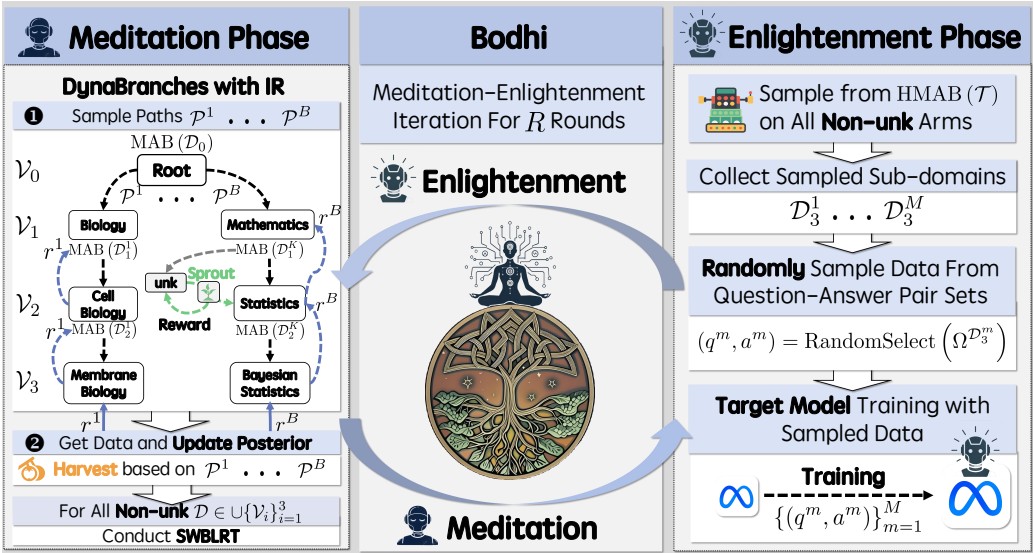

Figure 3: Schematic Diagram of the Iterative Process of Bohdi.

❶ For each arm $\mathcal{D}$ in the arm set $\mathcal{C}(\mathcal{D}_i^b, \mathcal{T})$ of the MAB instantiated by the current domain $\mathcal{D}_i^b$, sample the reward distribution parameter $\lambda_\mathcal{D} \sim \text{Beta}(\alpha_\mathcal{D}, \beta_\mathcal{D})$ from its posterior $\text{Beta}(\alpha_\mathcal{D}, \beta_\mathcal{D})$, and then select the arm with the maximum observed reward distribution parameter $\mathcal{D} = \arg\max_{\mathcal{D} \in \mathcal{C}(\mathcal{D}_i^b, \mathcal{T})} \lambda_\mathcal{D}$ as the candidate arm.

❷ For the candidate arm $\mathcal{D}$, if it is not an unk arm, then directly add it as the new domain $\mathcal{D}_{i+1}^b = \mathcal{D}$ to the path $\mathcal{P}^b$ and $\mathcal{V}_{i+1}$. Otherwise, perform the Sprout operation to obtain $\mathcal{D}_{\exp} = \text{Sprout}(\mathcal{D}_i^b, \mathcal{S}, \mathcal{V})$. If the returned $\mathcal{D}_{\exp}$ is non-empty, then add it as the new domain $\mathcal{D}_{i+1}^b = \mathcal{D}_{\exp}$ to $\mathcal{P}^b$ and $\mathcal{V}_{i+1}$, then update the posterior parameter of the unk arm $\mathcal{D}$ with $\alpha_\mathcal{D} = \alpha_\mathcal{D} + 1$. If $\mathcal{D}_{\exp}$ is empty, then repeat step ❶ to resample on MAB $(\mathcal{D}_i^b)$ and update the posterior parameter of the unk arm $\mathcal{D}$ with $\beta_\mathcal{D} = \beta_\mathcal{D} + 1$.

*Step2: Parallel Posterior Update.* In this step, we perform the Harvest operation on the sampled paths $\{\mathcal{P}^b\}_{b=1}^B$ to generate question-answer pairs and add them to the corresponding question-answer pair set $\Omega^{\mathcal{D}_3^b}$. Meanwhile, we calculate the reward based on the model's response and update the parameters of the sampled arms in $\mathcal{P}^b$.

For detailed descriptions of the two steps mentioned above, please refer to Appendix B.2.

**IR Mechanism for Online Sampling** To further enable the sampling process to adapt online to changes in the reward distribution caused by target LLM updates and to avoid over-reliance on old observed rewards, we propose the IR mechanism. It dynamically checks the consistency between the observed reward distribution within a sliding window and the overall empirical reward distribution via the designed SWBLRT. When a change in distribution is detected, the reward distribution parameters of the corresponding arms are reset to reduce empirical dependence on old observations.

Specifically, for each non-unk arm $\mathcal{D}$, we initialize a sliding window of width $w$ to collect the most recent $w$ reward observations. Assuming that there are currently $n$ observed rewards $\{r_\mathcal{D}^i\}_{i=1}^n$ on $\mathcal{D}$, if $n \leq w$, sampling and updating proceed according to DynaBranches. If $n > w$, we construct an SWBLRT to test the samples within the sliding window $\{r_\mathcal{D}^i\}_{i=n-w+1}^n$, with the null hypothesis and the alternative hypothesis are $\mathcal{H}_0 : \sum_{i=n-w+1}^n r_\mathcal{D}^i \sim \text{Binomial}(w, \lambda_\mathcal{D}^{1:n})$ and $\mathcal{H}_1 : \sum_{i=n-w+1}^n r_\mathcal{D}^i \not\sim \text{Binomial}(w, \lambda_\mathcal{D}^{1:n})$ respectively, where $\lambda_\mathcal{D}^{n_1:n_2} = \frac{\sum_{i=n_1}^{n_2} r_\mathcal{D}^i}{n_2 - n_1 + 1}$ denotes the probability parameter of the binomial distribution for the observed rewards from the $n_1$-th to the $n_2$-th. Let $L(\mathcal{H}_0)$ and $L(\mathcal{H}_1)$ denote the likelihood functions of the observed rewards $\{r_\mathcal{D}^i\}_{i=n-w+1}^n$ under $\mathcal{H}_0$ and $\mathcal{H}_1$ respectively. Then the test statistic for the SWBLRT is $\Lambda = -2\log\frac{L(\mathcal{H}_0)}{L(\mathcal{H}_1)}$ [12], with its specific form and derivation under our settings detailed in Appendix B.3.

Table 1: Results of the comparative experiments (accuracy in %). The best results and the second-best results are highlighted in bold and underlined, respectively. The numbers after the arrows indicate the absolute improvement (shown in red) or decline (shown in green) compared with the original model.

| Model | Data Volume | Multidisciplinary | | Mathematic | | Programming | | Reasoning | | AVG |
|---|---|---|---|---|---|---|---|---|---|---|
| | | MMLU (5-Shot) | GPQA (0-Shot CoT) | GSM8K (0-Shot CoT) | MATH (0-Shot CoT) | HumanEval (0-Shot) | MBPP (3-Shot) | TheoremQA (5-Shot) | BBH (3-Shot CoT) | |
| **Source Models** | | | | | | | | | | |
| Q-14B | – | 78.54 | 45.45 | 90.22 | 75.04 | 78.66 | 70.60 | 22.88 | 80.45 | 67.73 |
| M-24B | – | 80.95 | 46.46 | 91.89 | 68.84 | 79.88 | 68.20 | 37.12 | 82.71 | 69.51 |
| P-14B | – | 81.26 | 52.53 | 86.88 | 74.66 | 84.15 | 71.20 | 37.38 | 82.09 | 71.27 |
| **Target Model: Llama3.2-3B-Instruct** | | | | | | | | | | |
| Base | – | 59.77 | 27.78 | 68.92 | 34.82 | 56.10 | 48.20 | 23.88 | 53.86 | 46.67 |
| FuseChat | 95K | $59.88_{\uparrow 0.11}$ | $\underline{28.28}_{\uparrow 0.50}$ | $69.52_{\uparrow 0.60}$ | $36.14_{\uparrow 1.32}$ | $54.27_{\downarrow 1.83}$ | $47.60_{\downarrow 0.60}$ | $\underline{24.25}_{\uparrow 0.37}$ | $\mathbf{55.03}_{\uparrow 1.17}$ | $46.87_{\uparrow 0.20}$ |
| SFT | 90K | $\mathbf{62.32}_{\uparrow 2.55}$ | $25.76_{\downarrow 2.02}$ | $76.65_{\uparrow 7.73}$ | $48.92_{\uparrow 14.1}$ | $55.49_{\downarrow 0.61}$ | $\underline{50.60}_{\uparrow 2.40}$ | $14.75_{\downarrow 9.13}$ | $46.08_{\downarrow 7.78}$ | $47.57_{\uparrow 0.90}$ |
| FuseChat-3.0 | 90K | $\underline{62.03}_{\uparrow 2.26}$ | $22.22_{\downarrow 2.02}$ | $\underline{78.39}_{\uparrow 7.73}$ | $\mathbf{52.42}_{\uparrow 17.6}$ | $\underline{57.93}_{\uparrow 1.83}$ | $\mathbf{51.00}_{\uparrow 2.80}$ | $18.62_{\downarrow 5.26}$ | $44.97_{\downarrow 8.89}$ | $\underline{48.45}_{\uparrow 1.78}$ |
| Condor | 90K | $60.50_{\uparrow 0.73}$ | $27.78_{\uparrow 0.00}$ | $71.57_{\uparrow 2.65}$ | $\underline{48.96}_{\uparrow 14.14}$ | $53.66_{\downarrow 2.44}$ | $45.80_{\downarrow 2.40}$ | $22.38_{\downarrow 1.50}$ | $44.01_{\downarrow 9.85}$ | $46.83_{\uparrow 0.16}$ |
| Bohdi (Ours) | 1733 | $61.33_{\uparrow 1.56}$ | $\mathbf{30.30}_{\uparrow 2.52}$ | $\mathbf{78.62}_{\uparrow 9.70}$ | $47.92_{\uparrow 13.10}$ | $\mathbf{58.54}_{\uparrow 2.44}$ | $49.60_{\uparrow 1.40}$ | $\mathbf{25.12}_{\uparrow 1.24}$ | $\underline{54.90}_{\uparrow 1.04}$ | $\mathbf{50.79}_{\uparrow 4.12}$ |
| **Target Model: Gemma2-9B-IT** | | | | | | | | | | |
| Base | – | 71.59 | 28.79 | 81.50 | 45.96 | 65.85 | 57.60 | 15.62 | 70.97 | 54.74 |
| FuseChat | 95K | $\underline{71.76}_{\uparrow 0.17}$ | $34.34_{\uparrow 5.55}$ | $\underline{80.14}_{\downarrow 1.36}$ | $46.88_{\uparrow 0.92}$ | $\mathbf{66.46}_{\uparrow 0.61}$ | $\underline{58.80}_{\uparrow 1.20}$ | $16.62_{\uparrow 1.00}$ | $71.42_{\uparrow 0.45}$ | $\underline{55.80}_{\uparrow 1.06}$ |
| SFT | 90K | $60.50_{\downarrow 11.09}$ | $30.81_{\uparrow 2.02}$ | $68.43_{\downarrow 13.07}$ | $50.00_{\uparrow 4.04}$ | $64.02_{\downarrow 1.83}$ | $56.40_{\downarrow 1.20}$ | $15.38_{\downarrow 0.24}$ | $71.60_{\uparrow 0.63}$ | $52.14_{\downarrow 2.60}$ |
| FuseChat-3.0 | 90K | $66.12_{\downarrow 5.47}$ | $30.81_{\uparrow 2.02}$ | $70.35_{\downarrow 11.15}$ | $47.54_{\uparrow 1.58}$ | $62.20_{\downarrow 3.65}$ | $57.20_{\downarrow 0.4}$ | $\underline{20.00}_{\uparrow 4.38}$ | $71.08_{\uparrow 0.11}$ | $53.16_{\downarrow 1.58}$ |
| Condor | 90K | $70.03_{\downarrow 1.56}$ | $\underline{36.87}_{\uparrow 8.08}$ | $68.61_{\downarrow 12.89}$ | $\underline{51.08}_{\uparrow 5.12}$ | $\underline{65.85}_{\uparrow 0.00}$ | $57.80_{\uparrow 0.20}$ | $14.88_{\downarrow 0.74}$ | $\mathbf{74.46}_{\uparrow 3.49}$ | $54.95_{\uparrow 0.21}$ |
| Bohdi (Ours) | 1756 | $\mathbf{71.77}_{\uparrow 0.18}$ | $\mathbf{37.88}_{\uparrow 9.09}$ | $\mathbf{83.62}_{\uparrow 2.12}$ | $\mathbf{54.20}_{\uparrow 8.24}$ | $64.02_{\downarrow 1.83}$ | $\mathbf{60.40}_{\uparrow 2.80}$ | $\mathbf{26.88}_{\uparrow 11.26}$ | $\underline{72.40}_{\uparrow 1.43}$ | $\mathbf{58.90}_{\uparrow 4.16}$ |

According to Wilks' theorem [13], the test statistic $\Lambda$ asymptotically follows a chi-squared distribution with 1 degree of freedom, $\chi_1^2$. Therefore, we construct the rejection region $\left\{\Lambda > \chi_{1,1-u}^2\right\}$, where $\chi_{1,1-u}^2$ is the $(1-u)$-th quantile of $\chi_1^2$. When $\Lambda$ falls into the rejection region, we reject the null hypothesis $\mathcal{H}_0$, indicating a significant difference between the reward distribution within the sliding window and the historical empirical distribution, and the current posterior of arm $\mathcal{D}$ no longer accurately reflects its true reward characteristics. To address this, we reset the posterior of arm $\mathcal{D}$ to $\text{Beta}\left(\sum_{i=n-w+1}^{n} r_{\mathcal{D}}^i, w - \sum_{i=n-w+1}^{n} r_{\mathcal{D}}^i\right)$ based on the observed rewards within the sliding window, in order to eliminate the reward distribution bias brought by outdated observed rewards. By integrating the proposed IR mechanism, DynaBranches can adapt online to changes in the reward distribution and maintain sensitivity to the most recent observational data.

Combining the two mechanisms described above, Bohdi samples $B$ paths on HMAB($\mathcal{T}$) and updates the MAB parameters using DynaBranches incorporating the IR mechanism in the Meditation phase. During the Enlightenment phase, we conduct sampling solely on HMAB($\mathcal{T}$), while disabling the Sprout operation, i.e., sampling is restricted to non-unk arms, then we randomly sample from all existing question-answer pairs under each sampled path to obtain $M$ data for training $\mathcal{M}^T$. In each round, the Meditation and Enlightenment phases are executed sequentially until the $R$-th round is completed, as shown in Fig. 3. The corresponding overall pseudocode is provided in Appendix B.4.

## 3 Experiments

To validate the effectiveness of Bohdi, we select a wide range of models with different architectures and sizes as source and target models. The source models include Qwen2.5-14B-Instruct (denoted as Q-14B) [1], Mistral-Small-24B-Instruct-2501 (denoted as M-24B) [14], and phi-4 (denoted as P-14B) [15]. We choose two LLMs with different architectures and sizes as target models: Llama3.2-3B-Instruct [16] and Gemma2-9B-IT [17]. Additional experimental results for more target models can be found in Appendix D.

### 3.1 Experimental Setup

We provide a general introduction to the experimental setup. For detailed experimental settings, including training and evaluation configurations, please refer to Appendix C.

**Benchmarks for Evaluation** To more comprehensively evaluate the performance improvements of the fused target model across different dimensions, we select two benchmarks for each of the four main capability dimensions: multidisciplinary knowledge, mathematics, programming, and reasoning. For multidisciplinary knowledge, we select the multidisciplinary question-answering benchmark MMLU [18] and the natural science subject question-answering benchmark GPQA [19]. For mathematics, we choose the mathematical problem-solving benchmarks GSM8K [20] and MATH

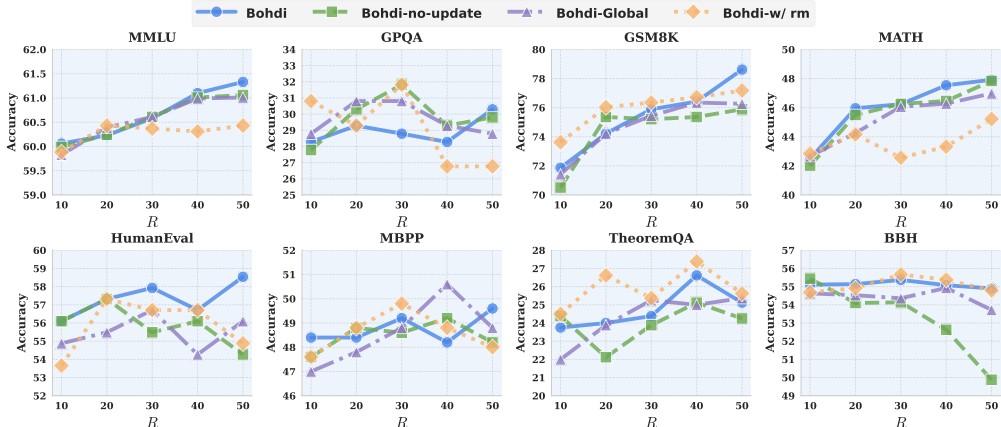

Figure 4: Comparison of Bohdi under various component setting across different fusion rounds $R$.

[21]. For programming, we select the programming benchmarks HumanEval [22] and MBPP [23]. For reasoning ability, we choose the benchmark BBH [24] for measuring logical reasoning ability and the theorem-driven reasoning benchmark TheoremQA [25]. To ensure a unified and fair evaluation, we use Opencompass [26] as the evaluation suite.

**Baselines** We compare the target models fused using Bohdi with target models trained using several other approaches. These include fused models obtained through the most representative EF method FuseChat [6] and the state-of-the-art IF method FuseChat-3.0 [7, 9]. As a control, we also consider target models that are Supervised Fine-Tuned (SFT) directly on a given dataset by collecting responses from each source model, as well as those that use Condor [27] to generate questions and answers for multiple domains based on each source model and then perform SFT on the target model for knowledge fusion.

**Dataset and Training Settings** For FuseChat, we use the FuseChat-Mixture dataset, which consists of 95,000 dialogue data as used in its original paper [6]. For SFT and FuseChat3.0, we collect 10,000 dialogues each in mathematics, programming, and general domains, totaling 30,000 dialogues, and gather responses from the three source models, resulting in a dataset of 90,000 data for training the target model. For FuseChat3.0, following the original paper [7], we use ArmoRM-Llama3-8B-v0.1 [28] as the reward model. To ensure a similar data volume, for Condor, we use the provided instructions [27] to construct 30,000 dialogues based on each source model, totaling 90,000 data. For Bohdi, in the comparative experiments, we perform $R = 50$ iterations, sampling $B = 90$ paths in the Meditation phase and $M = 180$ data in the Enlightenment phase for training the target model in each round, with the quantile parameter $u = 0.2$ and window width $w = 20$ for SWBLRT.

## 3.2 Main Results

The comparative results are reported in Tab. 1. For each target model, the average performance after fusion with Bohdi surpasses that of all other baseline methods, achieving average performance improvements of $4.12\%$ for Llama3.2-3B-Instruct and $4.16\%$ for Gemma2-9B-IT. Benefiting from Bohdi's comprehensive knowledge domain expansion and automatic domain allocation capabilities, the target models fused with Bohdi demonstrate improved average performance without significant degradation in any specific capabilities, achieving either optimal or sub-optimal performance on most benchmarks. Notably, Bohdi provides consistent performance improvements across all benchmarks for Llama3.2-3B-Instruct. Moreover, Bohdi enhances the performance of Gemma2-9B-IT on TheoremQA from $15.62$ to $26.88$, a performance improvement of over $72\%$, surpassing the larger Q-14B. In contrast, although FuseChat-3.0 and SFT can achieve the highest performance improvements on MMLU and MATH for the original Llama3.2-3B-Instruct, they also cause severe performance drops (up to over $30\%$) on TheoremQA and BBH, leading to a significant capability imbalance. Additionally, in Tab. 1, we also report the amount of data used for training by each method. Bohdi's training process uses only 1.7K data, which is approximately $1.8\%$ of the data used by other methods, further demonstrating Bohdi's advantage in data efficiency.

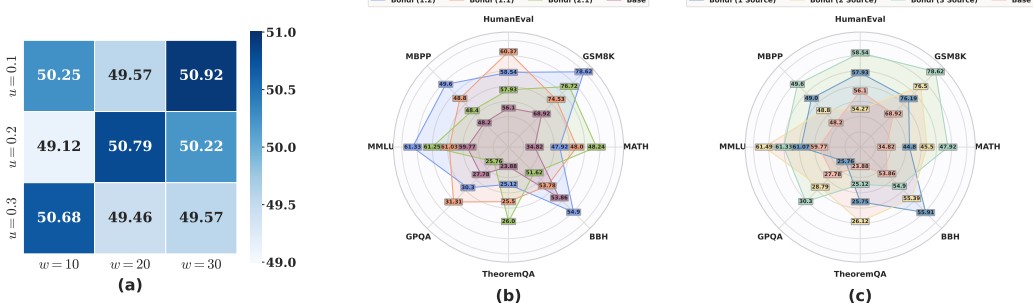

Figure 5: Results of various ablation studies. (a): The average performance of Bohdi across benchmarks under different settings of the quantile parameter $u$ and the window width parameter $w$. (b): Ablation results under different Meditation-Enlightenment sampling quantity ratios. (c): Ablation results under different number of source models $K$.

## 3.3 Ablation Studies

We conduct extensive ablation studies to verify the impact of various components and parameter settings on Bohdi. Unless otherwise stated, the settings in the experiments below are the same as those in the main experiments. Additional ablations and results are provided in Appendix D.

**Component Ablation** To verify the impact of DynaBranches and the IR mechanism on Bohdi's performance, we compare Bohdi without DynaBranches (performing only domain exploration and using uniform probability sampling for all known domains, denoted as **Bohdi-no-update**), Bohdi using DynaBranches but without the IR mechanism (with SWBLRT window size $w = \infty$ to disable IR, denoted as **Bohdi-Global**), and Bohdi with both mechanisms combined. We evaluate these variants on Llama3.2-3B-Instruct across different number of fusion rounds $R = \{10, 20, 30, 40, 50\}$. The results shown in Fig. 4 indicate that Bohdi with both mechanisms exhibits the most stable growth trend in capabilities across all benchmarks. In contrast, **Bohdi-Global** shows more fluctuation in its growth curves across benchmarks, demonstrating that the IR mechanism enables Bohdi to more stably track the target model's capability changes. **Bohdi-no-update**, on the other hand, experiences noticeable performance degradation on HumanEval and BBH, further highlighting that the design of DynaBranches can better avoid the capability imbalance through data allocation adjustments. Additionally, we are curious whether evaluating responses using a Reward Model is more effective than using a Leader Model. To explore this, we use the same reward model as FuseChat-3.0, ArmoRM-Llama3-8B-v0.1, to replace the Leader Model for response evaluation (denoted as **Bohdi w/ rm**). The results are also documented in Fig. 4. Due to the limited cross-domain generalization ability of the reward model [29], **Bohdi w/ rm** exhibits more pronounced performance fluctuations compared to Bohdi evaluated with the Leader Model, especially in GPQA and MBPP.

**Parameter Ablation for SWBLRT** With Llama3.2-3B-Instruct as the target model, we conduct ablation studies on the quantile parameter $u = \{0.1, 0.2, 0.3\}$ and the window width parameter $w = \{10, 20, 30\}$ in the IR mechanism, reporting the average performance across each benchmarks. The results shown in Fig. 5 (a) indicate that Bohdi achieves relatively higher performance at $u = 0.1, w = 30$, $u = 0.2, w = 20$, and $u = 0.3, w = 10$. This suggests that stricter detection criteria (lower $u$ values) require a longer window to enhance stability, while more lenient criteria benefit from a shorter window to speed up the update of reward distribution parameters.

**Ablation of Sampling Quantities for Each Phase** We evaluate the performance of the target model Llama3.2-3B-Instruct on various benchmarks before and after fusion with Bohdi under different sampling ratios between the Meditation ($B$) and Enlightenment ($M$) phases: $B : M = 1 : 2$, $B : M = 1 : 1$, and $B : M = 2 : 1$. The results are shown in in Fig. 5 (b). The fused model achieves the best average performance across dimensions when $B : M = 1 : 2$, obtaining the best performance in four dimensions. In contrast, when $B : M = 2 : 1$, the overall capability improvement is minimal. This is because a larger $B : M$ ratio generates excessive new data that cannot be sufficiently learned during the Enlightenment phase, limiting effective knowledge injection into the target model.

**Ablation on the Number of Source Models** We compare the performance of the target model Llama3.2-3B-Instruct fused using Bohdi on each benchmark when using $K = 1$ source model (only Q-14B), $K = 2$ source models (Q-14B and M-24B), and all three source models for fusion. The

Table 2: Experimental results of fusing the target model Gemma2-9B-IT with three smaller source models, including Llama3.2-3B-Instruct, Qwen2.5-3B-Instruct, and Gemma2-2B-IT using Bohdi. The numbers after the arrows indicate the absolute improvement (shown in red) or decline (shown in green) compared with the original model.

| Model | Multidisciplinary | | Mathematic | | Programming | | Reasoning | | AVG |
|---|---|---|---|---|---|---|---|---|---|
| | MMLU (5-Shot) | GPQA (0-Shot CoT) | GSM8K (0-Shot CoT) | MATH (0-Shot CoT) | HumanEval (0-Shot) | MBPP (3-Shot) | TheoremQA (5-Shot) | BBH (3-Shot CoT) | |
| Llama3.2-3B-Instruct | 59.77 | 27.78 | 68.92 | 34.82 | 56.10 | 48.20 | 23.88 | 53.86 | 46.67 |
| Qwen2.5-3B-Instruct | 65.33 | 37.88 | 82.34 | 62.96 | 73.17 | 56.80 | 15.00 | 46.18 | 54.95 |
| Gemma2-2B-IT | 56.69 | 20.20 | 59.21 | 18.68 | 45.73 | 41.00 | 7.25 | 42.52 | 36.41 |
| Gemma2-9B-IT | 71.59 | 28.79 | 81.50 | 45.96 | 65.85 | 57.60 | 15.62 | 70.97 | 54.74 |
| Bohdi-Gemma2-9B-IT | $71.46_{\downarrow 0.13}$ | $35.86_{\uparrow 7.07}$ | $81.50_{\uparrow 0.00}$ | $48.80_{\uparrow 2.84}$ | $62.20_{\downarrow 3.65}$ | $59.40_{\uparrow 1.80}$ | $19.25_{\uparrow 3.63}$ | $71.22_{\uparrow 0.25}$ | $56.21_{\uparrow 1.47}$ |

results are shown in Fig. 5 (c). Additionally, in Fig. 3.3, we display the number of domains at each level proposed by Bohdi during the entire 50 rounds of fusion when using $K = 1, 2, 3$ source models. As $K$ increases, the number of domains of each level proposed during fusion almost consistently rises, and the model tends to exhibit more pronounced growth in individual capabilities. This also indicates that involving more source models can introduce more comprehensive knowledge and bring about further improvements in the target model's capabilities.

**Attempts at Weak-To-Strong Supervision** To test Bohdi's potential as a weak-to-strong supervision paradigm, we use the largest of several target models, Gemma2-9B-IT, as the target model and employ three models with significantly smaller parameter sizes, including Llama3.2-3B-Instruct, Qwen2.5-3B-Instruct [1], and Gemma2-2B-IT [17], as source models for fusion using Bohdi (the fused model is denoted as Bohdi-Gemma2-9B-IT). The corresponding results are documented in Tab. 2. The results indicate that Bohdi still holds promise for enhancing the target model's capabilities, even when the source models are much smaller in size compared to the target model. Specifically, even when the source models are slightly smaller and overall weaker than the target model, the target model can still absorb the strengths of the source models in areas where

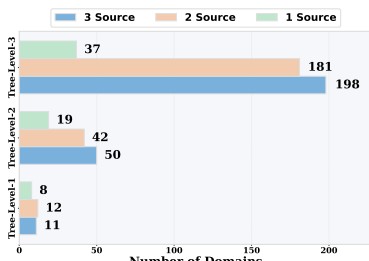

Figure 6: The number of proposed domains at each level during Bohdi's fusion process for different numbers of source models $K$.

it was originally weaker. Notably, the target model Gemma2-9B-IT is outperformed by the source model Qwen2.5-3B-Instruct on GPQA, MATH, and TheoremQA. However, the fused model Bohdi-Gemma2-9B-IT shows improved performance on these three benchmarks, while maintaining stable performance in other domains, resulting in a absolute average performance improvement of $1.47\%$. This demonstrates Bohdi's potential as a weak-to-strong supervision paradigm.

## 4  Conclusion

In this paper, we introduce Bohdi, a heterogeneous LLM fusion framework that organizes knowledge domains into a hierarchical tree structure, and formalizes both novel domain exploration and cross-domain data proportion allocation on this knowledge tree as a HMAB problem. We then propose the DynaBranches mechanism for dynamic sampling on the HMAB. This mechanism integrates the designed Sprout and Harvest operations to enable automated dynamic domain exploration, data acquisition, and adaptive adjustment of data proportions tailored to the target LLM's capabilities through multi-model collaboration. Furthermore, we develop the IR mechanism, empowering DynaBranches to dynamically track the target LLM's capability evolution and perform online adjustment of domain data proportions via the proposed SWBLRT. By orchestrating these components into a two-phase iterative process of Meditation and Enlightenment, Bohdi achieves automated knowledge boundary expansion and source LLM knowledge acquisition without relying on real data, while effectively mitigating the imbalance in target LLM's capabilities during fusion. The Bohdi's design philosophy also holds promise for providing new research ideas for multi-model capability synergy enhancement and the development of weak-to-strong supervision paradigms.

## 5  Acknowledgements

This work is supported by the Shanghai Municipal Science and Technology Major Project.

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

# A  Related Works

## A.1  Heterogeneous LLM Fusion

Heterogeneous LLM fusion aims to inject the knowledge from multiple source LLMs into a target LLM that serves as a pivot, thereby integrating the strengths of multiple powerful LLMs into a single LLM. Given that heterogeneous LLMs typically exhibit differences in architectures, parameter scales, and tokenizer vocabularies, current research primarily employs two dominant strategies to address these constraints: EF [5, 6, 30] and IF [7, 8, 9].

EF methods perform supervised distillation of the target LLM by collecting output probability distributions from source LLMs on a given dataset to explicitly transfer knowledge from multiple source LLMs to the target LLM. Due to misaligned output probability distributions caused by different vocabularies, EF methods employ various approaches to align these distributions across vocabularies. For example, aligning vocabularies by minimizing the edit distance between the output sequences of the source and target LLMs [5, 6], while others directly align the tokens with the highest prediction probabilities by minimizing the 1-Wasserstein distance [30]. In terms of fusion approaches, EF methods either uniformly align the target LLM with all source LLMs' probability distributions [5, 30] or first perform pairwise alignment, followed by parameter merging of the aligned target LLMs to obtain a final fused target LLM, thereby more stably integrating knowledge from multiple source LLMs. However, the vocabulary alignment process inevitably introduces alignment errors, which adds noise to the fusion process of EF.

In contrast, IF methods train the target LLM directly using responses from source LLMs, bypassing the vocabulary alignment process and thus enabling more stable knowledge injection. Their fusion approaches include selecting the best response from multiple source LLMs based on scores from an external reward LLM to guide the target LLM's preference optimization [8], or a two-phase process involving weighted SFT and DPO based on source LLM responses and reward LLM scores to more comprehensively utilize all responses from multiple source LLMs [7, 9].

Although current heterogeneous fusion methods have shown promising performance, they rely on the given data for the fusion process. Since comprehensive and systematic datasets across diverse domains are difficult to obtain, these methods can only rely on open-source data from a few domains for LLM alignment, which prevents the full injection of source LLMs' knowledge into the target LLM. Moreover, these methods use fixed data proportions, which cannot adaptively adjust the data proportions for each domain based on the target LLM's capabilities, leading to a imbalance in the target LLM's capabilities across different domains.

## A.2  LLM Data Synthesis

With the rapid increase in the parameter size of LLMs and the discovery of Scaling Laws [31, 32], the importance of data for the development of LLM capabilities has gradually become a consensus. Considering the high costs associated with data collection, organization, and annotation, using LLMs to generate synthetic data has emerged as a more cost-effective and efficient approach. The primary synthetic data generation schemes can be divided into seed-based synthesis and seedless synthesis.

Seed-based data synthesis involves augmenting or iteratively refining instructions and responses based on partially real, given instance data. For example, guiding the LLM to generate instruction and response data for various tasks based on carefully crafted task seed sets [33], synthesizing deeper and more diverse instructions by performing deep and broad evolution based on an initial set of instructions [34], iteratively enhancing initial questions and adding additional reasoning steps [35], and generating the logic to answer a large number of questions from a few examples, then detecting and iteratively improving responses to unresolved problems [36].

In contrast, seedless synthesis does not rely on external real data examples to generate data from scratch, thereby further avoiding the collection and annotation costs of pre-given real data. Examples include using LLMs to generate concepts, objects, and entities present in the real world, and creating corresponding questions and dialogues for each sub-topic [37]; using hand-written meta-prompts to have LLMs automatically generate multiple instructions and randomly select them to construct diverse instruction sets [38]; guiding LLMs to generate instructions and their corresponding responses using the response templates of Chat LLMs [39]; and generating dialogue data in multiple scenarios through

large-scale agent systems playing different roles [40]. However, since the aforementioned approaches either rely on closed-source models [37], or fail to cover extensive and diverse domains [38, 39], or require massive agent collaboration with intricate framework design thus necessitating cumbersome preparatory steps [40], Condor [27] organizes world knowledge through a tree structure, employs LLMs to generate comprehensive hierarchical domain labels, and further produces domain-specific QA data based on these labels.

Yet during each domain expansion, Condor generates all child domain labels at once, lacking autonomous exploration capability for new domains at various levels. Condor does not truly assign node information to each domain, failing to manage the diverse generated data through a genuinely tree-structured organizational scheme, and consequently cannot design feedback-based data ratio adjustment mechanisms across different domain levels.

# B Details of Bohdi Framework

## B.1 Details of Operations Sprout and Harvest

**Details of Operation Sprout** For domain $\mathcal{D}$ at level $i$ ($0 \leq i \leq 2$, when $i = 3$, further domain expansion is not necessary), if the selected child domain $\mathcal{D}_{i+1,j}$ is an unk node $\mathcal{D}_{i+1,\mathrm{unk}}$, all source LLMs collaboratively propose a new domain $\mathcal{D}^{\mathrm{new}} = \mathrm{Sprout}(\mathcal{D}, \mathcal{S}, \mathcal{V})$ for domain expansion. Specifically, for the already sampled set of parent domains $\mathrm{Parent}(\mathcal{D}_{i,j})$, we construct the expansion prompt $p\left[\mathrm{Expand}, \mathrm{Parent}\left(\mathcal{D}_{i,j}\right)\right]$ (see Appendix F for the specific format), which is answered by all source LLMs $\mathcal{M}_k^S$ to obtain proposed new domains $\mathcal{D}_k^{\mathrm{new}} = \mathcal{M}_k^S\left(p\left[\mathrm{Expand}; \mathrm{Parent}\left(\mathcal{D}_{i,j}\right)\right], \boldsymbol{\theta}_k^S\right)$. For the proposed domains $\{\mathcal{D}_k^{\mathrm{new}}\}_{k=1}^K$, we first exclude any existing domains contained within them, and then perform a majority vote among them to elect the new domain: $\mathcal{D}^{\mathrm{New}} = \mathrm{Elect}\left(\{\mathcal{D}_k^{\mathrm{new}}\}_{k=1}^K - \cup_{j=1}^3 \mathcal{V}_j\right)$, then add it to the node set $\mathcal{V}_i$ of the corresponding level and update $N_i = N_i + 1$. If $\{\mathcal{D}_k^{\mathrm{new}}\}_{k=1}^K - \cup_{j=1}^3 \mathcal{V}_j = \emptyset$, then this node expansion will be discarded. If all proposed domains are different, the election operation degenerates to randomly selecting one from the candidate domains as $\mathcal{D}^{\mathrm{new}}$.

**Details of Operation Harvest** Given the sampled path $\mathcal{P} = [\mathcal{D}_{0,0} \to \mathcal{D}_{1,i} \to \mathcal{D}_{2,j} \to \mathcal{D}_{3,k}]$, Harvest operation leverages the given LLMs to generate question-answer pairs $(q, a) = \mathrm{Harvest}(\mathcal{P}, \mathcal{S}, \mathcal{M}^T)$ belonging to the subdomain $\mathcal{D}_{3,k}$ on $\mathcal{P}$ to obtain the sampled data. Specifically, we first randomly assign a Leader LLM $\mathcal{M}^{\mathrm{Leader}} \in \{\mathcal{M}_k^S\}_{k=1}^K$ from the set of source LLMs to generate a question $q = \mathcal{M}^{\mathrm{Leader}}(p\left[\mathrm{Inquiry}; \mathcal{P}\right]) \in \mathcal{Q}^{\mathcal{D}_{3,k}}$ based on the sampled path, where $p\left[\mathrm{Inquiry}; \mathcal{P}\right]$ is the prompt that guides the LLM to generate a question corresponding to the given path $\mathcal{P}$. To further obtain answers and calculate $V(q, a)$, we collect a set of answers $\mathcal{A}^q = \left\{a_i \mid a_i = \mathcal{M}_i(q, \boldsymbol{\theta}_i), \mathcal{M}_i \in \{\mathcal{M}_k^S\}_{k=1}^K \cup \{\mathcal{M}^T\}\right\}$ from all LLMs for question $q$, and use the Leader LLM to judge and select the best answer: $a^{q,\mathrm{Best}} = \mathcal{M}^{\mathrm{Leader}}\left(p\left[\mathrm{Select}; q, \mathcal{A}^q\right], \boldsymbol{\theta}^{\mathrm{Leader}}\right)$, where $p[\mathrm{Select}; q, \mathcal{A}^q]$ represents the prompt that guides the Leader LLM in selecting the best answer. The specific formats of related prompts can be found in Appendix F. For any $a \in \mathcal{A}^q$, we define $V(q, a) = 1$ if and only if $a = a^{q,\mathrm{Best}}$, and $V(q, a) = 0$ otherwise. We then add the question-answer pair $(q, a^{q,\mathrm{Best}})$ to the sampled question-answer pair set $\Omega^{\mathcal{D}_{3,k}}$ of the corresponding subdomain $\mathcal{D}_{3,k}$ for target LLM training in the Enlightenment phase.

## B.2 Details of the DynaBranches

Here we provide the detailed descriptions of the two steps of DynaBranches.

*Step1: Batchwise Path Sampling.* In this phase, we sample $B$ paths $\{\mathcal{P}^b\}_{b=1}^B$ on $\mathrm{HMAB}(\mathcal{T})$, where $B$ is the batch size. For each path $\mathcal{P}^b$, we first initialize $\mathcal{P}^b = [\mathcal{D}_{0,0}]$ and sample the reward distribution parameters $\lambda_{\mathcal{D}^b}$ for each arm $\mathcal{D}^b \in \mathcal{C}(\mathcal{D}_{\mathrm{Now}}^b, \mathcal{T})$ on the first MAB $\mathrm{MAB}(\mathcal{D}_{0,0})$: $\lambda_{\mathcal{D}^b} \sim \mathrm{Beta}(\alpha_{\mathcal{D}^b}, \beta_{\mathcal{D}^b})$. We then select the arm with the highest sampled reward distribution parameter: $\mathcal{D} = \arg\max_{\mathcal{D} \in \mathcal{C}(\mathcal{D}^b, \mathcal{T})} \lambda_{\mathcal{D}}$. If the chosen arm $\mathcal{D} \in \mathcal{V}$ is not an unk arm, we directly add it as $\mathcal{D}_{i+1}^b$ to the path $\mathcal{P}^b$. If $\mathcal{D}$ is an unk arm $\mathcal{D}_{\mathrm{unk}}$, we perform the Sprout operation to expand an unknown arm: $\mathcal{D}_{\mathrm{exp}} = \mathrm{Sprout}(\mathcal{D}_{0,0}, \mathcal{S}, \mathcal{V})$. If $\mathcal{D}_{\mathrm{exp}}$ is a known arm, i.e., $\mathcal{D}_{\mathrm{exp}} \in \mathcal{V}$, the Sprout

operation is discarded, and a reward $r = 0$ is fed back to the arm $\mathcal{D}_{\text{unk}}$, then update $\beta_{\mathcal{D}_{\text{unk}}} = \beta_{\mathcal{D}_{\text{unk}}} + 1$ and resample an arm on $\text{MAB}(\mathcal{D}_{0,0})$. If the Sprout operation finds a new arm $\mathcal{D}_{\text{exp}} \notin \mathcal{V}$, we add it as $\mathcal{D}_{i+1}^b$ to the path $\mathcal{P}^b$, then feed back a reward $r = 1$ to the unk arm $\mathcal{D}_{\text{unk}}$, and update $\alpha_{\mathcal{D}_{\text{unk}}} = \alpha_{\mathcal{D}_{\text{unk}}} + 1$. After sampling $\mathcal{D}_{i+1}^b$, we continue sampling subsequent arms on the instantiated child-level MAB $\text{MAB}(\mathcal{D}_{i+1}^b)$ until the bottom-level MAB is sampled. To avoid too many invalid Sprout operations, if there are 10 invalid Sprout operations during the sampling process on the path $\mathcal{P}^b$, we mark the path $\mathcal{P}^b$ as "Invalid" and update the posterior distribution of the most recently sampled arm $\mathcal{D}_{\text{Now}}$ in the current $\mathcal{P}^b$ to $\text{Beta}(1, \infty)$ to prohibit sampling $\mathcal{D}_{\text{Now}}$ in subsequent sampling processes.

*Step2: Parallel Posterior Update.* For the sampled paths $\left\{ \mathcal{P}^b \right\}_{b=1}^B$, we perform the Harvest operation in parallel on all valid $\mathcal{P}^b$ to obtain question-answer pairs $(q^b, a^b) = \text{Harvest}(\mathcal{P}^b, \mathcal{S}, \mathcal{M}^T)$. Then calculate the reward $r^b = \mathbb{1}(a^b \neq \mathcal{M}^T(q^b, \boldsymbol{\theta}^T))$, and update the posterior of the reward distribution parameters for each arm $\mathcal{D}^b$ in $\mathcal{P}^b$ except for $\mathcal{D}_{0,0}$: $\alpha_{\mathcal{D}^b} = \alpha_{\mathcal{D}^b} + r^b$, $\beta_{\mathcal{D}^b} = \beta_{\mathcal{D}^b} + (1 - r^b)$.

### B.3 The formal construction of the SWBLRT test statistic

Since the sum of reward observations within the window $S^w = \sum_{i=n-w+1}^n r_{\mathcal{D}}^i$ follows a binomial distribution, its likelihood function $L(\lambda)$ can be expressed as

$$L(\lambda) = \binom{w}{S^w} \lambda^{S^w} (1 - \lambda)^{(w - S^w)}, \tag{7}$$

where $\lambda$ is the probability parameter of the binomial distribution $\text{Binomial}(w, \lambda)$ that $S^w$ follows. When the null hypothesis $\mathcal{H}_0 : \sum_{i=n-w+1}^n r_{\mathcal{D}}^i \sim \text{Binomial}(w, \lambda_{\mathcal{D}}^{1:n})$ holds, the likelihood function $L(\mathcal{H}_0)$ can be rewritten as:

$$L(\mathcal{H}_0) = \binom{w}{S^w} \left( \lambda_{\mathcal{D}}^{1:n} \right)^{S^w} \left( 1 - \lambda_{\mathcal{D}}^{1:n} \right)^{(w - S^w)}. \tag{8}$$

When the alternative hypothesis $\mathcal{H}_1 : \sum_{i=n-w+1}^n r_{\mathcal{D}}^i \not\sim \text{Binomial}(w, \lambda_{\mathcal{D}}^{1:n})$ holds, we consider its maximum likelihood. First, examine the derivative of the log-likelihood in Eq. (7):

$$\frac{\mathrm{d}L(\lambda)}{\mathrm{d}\lambda} = \frac{1}{\lambda} S^w - \frac{1}{1 - \lambda} (w - S^w), \tag{9}$$

let $\frac{\mathrm{d}L(\lambda)}{\mathrm{d}\lambda} = 0$, we obtain the maximum likelihood estimate of $\lambda$ as $\widehat{\lambda} = \frac{S^w}{w}$. Substituting this into Eq. (7) yields the likelihood function corresponding to the alternative hypothesis:

$$L(\mathcal{H}_1) = \binom{w}{S^w} \left( \frac{S^w}{w} \right)^{S^w} \left( 1 - \frac{S^w}{w} \right)^{w - S^w}. \tag{10}$$

Then the test statistic $\Lambda$ for SWBLRT can be derived as:

$$\Lambda = -2 \log \frac{L(\mathcal{H}_0)}{L(\mathcal{H}_1)} \tag{11}$$

$$= 2 S^w \log \left( \frac{\frac{S^w}{w}}{\lambda_{\mathcal{D}}^{1:n}} \right) + 2(w - S^w) \log \left( \frac{1 - \frac{S^w}{w}}{1 - \lambda_{\mathcal{D}}^{1:n}} \right)$$

$$= 2 \left[ \left( \sum_{i=n-w+1}^n r_{\mathcal{D}}^i \right) \log \left( \frac{\lambda_{\mathcal{D}}^{n-w+1:n}}{\lambda_{\mathcal{D}}^{1:n}} \right) + \left( w - \sum_{i=n-w+1}^n r_{\mathcal{D}}^i \right) \log \left( \frac{1 - \lambda_{\mathcal{D}}^{n-w+1:n}}{1 - \lambda_{\mathcal{D}}^{1:n}} \right) \right]. \tag{12}$$

Therefore, the test statistic used in the SWBLRT is calculated in the form shown in Eq. (12).

### B.4 Pseudocode of Bohdi

We present the pseudocode for Bohdi's actual execution process in Algorithm 1.

**Algorithm 1** The execution process of Bohdi

**Input** : Source LLMs $\{\mathcal{M}_k^S\}_{k=1}^K$, target LLM $\mathcal{M}^T$, number of paths sampled in the Meditation phase and the Enlightenment phase $B$ and $M$, the window width parameter $w$ and the quantile parameter $u$ of the IR mechanism, initialized knowledge tree $\mathcal{T}$ and HMAB $(\mathcal{T})$

**Output** : Fused target model $\mathcal{M}_{\text{Fuse}}^T$

**for** $t \leftarrow 1$ **to** $R$ **do**

    **MEDITATION PHASE**:

    *DynaBranches-Batchwise Sampling*: **for** $b \leftarrow 1$ **to** $B$ **do**

        $i \leftarrow 0, \mathcal{D}_0^b \leftarrow \mathcal{D}_{0,0}, \mathcal{P}^b \leftarrow [\mathcal{D}_{0,0}], \text{MAB}_i^b \leftarrow \text{MAB}(\mathcal{D}_{0,0}), \text{Fail} \leftarrow 0$

        **while** $i < 3$ *and* $\text{Fail} < 10$ **do**

            **parallel for** $\mathcal{D}^b \in \mathcal{C}(\mathcal{D}_i^b)$ **do**

                $\lambda_{\mathcal{D}^b} \sim \text{Beta}(\alpha_{\mathcal{D}^b}, \beta_{\mathcal{D}^b})$

            $\mathcal{D} \leftarrow \arg\max_{\mathcal{D} \in \mathcal{C}(\mathcal{D}_i^b)} \lambda_{\mathcal{D}}$          `// Sample an arm on MAB`$_i^b$

            **if** $\mathcal{D}$ *is unk* **then**

                $\mathcal{D}_{\text{exp}} \leftarrow \text{Sprout}(\mathcal{D}_i^b, \mathcal{S}, \mathcal{V})$

                **while** $\mathcal{D}_{\text{exp}}$ *is None and* $\text{Fail} < 10$ **do**

                    $\mathcal{D}_{\text{exp}} \leftarrow \text{Sprout}(\mathcal{D}_i^b, \mathcal{S}, \mathcal{V}), \text{Fail} \leftarrow \text{Fail} + 1, \beta_{\mathcal{D}} \leftarrow \beta_{\mathcal{D}} + 1$

                **if** $\text{Fail} < 10$ **then**

                    Initialize an unk domain $\mathcal{D}_{\text{unk}}$ as a child domain of $\mathcal{D}_{\text{exp}}$ with $\lambda_{\mathcal{D}_{\text{unk}}} \sim \text{Beta}(1, 1)$

                    Initialize the reward observation pool $\mathcal{W}_{\mathcal{D}_{\text{exp}}}$ for $\mathcal{D}_{\text{exp}}$

                    $\alpha_{\mathcal{D}} \leftarrow \alpha_{\mathcal{D}} + 1, \mathcal{D}_{i+1}^b \leftarrow \mathcal{D}_{\text{exp}}, \mathcal{V}_{i+1} \leftarrow \mathcal{V}_{i+1} \cup \{\mathcal{D}_{i+1}^b\}, \mathcal{E} \leftarrow \mathcal{E} \cup \{(\mathcal{D}_i^b, \mathcal{D}_{i+1}^b)\}$

                    **if** $i = 2$ **then**

                        Initialize the question-answer pair set $\Omega^{\mathcal{D}_{i+1}^b} = \emptyset$ for $\mathcal{D}_{i+1}^b$

            **else**

                $\mathcal{D}_{i+1}^b \leftarrow \mathcal{D}$

            $\mathcal{P}^b$ append $\mathcal{D}_{i+1}^b$          `// Append `$\mathcal{D}_{i+1}^b$` to the sampled path `$\mathcal{P}^b$

        **if** $\text{Fail} = 10$ **then**

            $(\alpha_{\mathcal{D}_i^b}, \beta_{\mathcal{D}_i^b}) \leftarrow (1, \infty)$, Mark $\mathcal{P}^b$ as Invalid

        **else**

            Initialize the question-answer pair set $\Omega^{\mathcal{D}_3^b}$ for $\mathcal{D}_3^b$

    *DynaBranches-Batchwise Posterior Update*: **for** *all valid* $\mathcal{P}^b \in \{\mathcal{P}^b\}_{b=1}^B$ **do**

        $(q^b, a^b) \leftarrow \text{Harvest}(\mathcal{P}^b, \mathcal{S}, \mathcal{M}^T), r^b \leftarrow \mathbb{1}(a^b \neq \mathcal{M}^T(q^b, \boldsymbol{\theta}))$

        **for** $\mathcal{D} \in \mathcal{P}^b \setminus \{\mathcal{D}_{0,0}\}$ **do**

            $\alpha_{\mathcal{D}} \leftarrow \alpha_{\mathcal{D}} + r^b, \beta_{\mathcal{D}} \leftarrow \beta_{\mathcal{D}} + (1 - r^b), \mathcal{W}_{\mathcal{D}}$ append $r^b$      `// Update Posterior`

            **if** $\mathcal{D} = \mathcal{D}_3^b$ **then**

                $\Omega^{\mathcal{D}_3^b} = \Omega^{\mathcal{D}_3^b} \cup \{(q^b, a^b)\}$

    *IR Mechanism with SWBLRT*: $\mathcal{D}_0 \leftarrow \mathcal{D}_{0,0}$, **for** $i \leftarrow 0$ **to** $2$ **do**

        **for** *all non-unk arm* $\mathcal{D}_{i+1} \in \mathcal{C}(\mathcal{D}_i)$ **do**

            **if** *the number of reward observations* $n_{\mathcal{D}_{i+1}}$ *in* $\mathcal{W}_{\mathcal{D}_{i+1}}$ *satisfies* $n_{\mathcal{D}_{i+1}} > w$ **then**

                Calculate the SWBLRT statistic $\Lambda_{\mathcal{D}_{i+1}}$ for $\mathcal{D}_{i+1}$

                **if** $\Lambda_{\mathcal{D}_{i+1}} > \chi_{1,1-u}^2$ **then**

                    Reset the reward distribution parameter posterior of arm $\mathcal{D}_{i+1}$ based on the latest $w$ observations in $\mathcal{W}_{\mathcal{D}_{i+1}}$

    **ENLIGHTENMENT PHASE**:                               `// Target Model Training`

    Sample $M$ paths $\{\mathcal{P}^m\}_{m=1}^M$ from all non-unk arms in $\text{HMAB}(\mathcal{T})$ without updating the posteriors

    Initialize $\mathcal{B} = \emptyset$

    **parallel for** $m \leftarrow 1$ **to** $M$ **do**

        Randomly select a question-answer pair $(q^m, a^m) \in \Omega^{\mathcal{D}_3^m}$ from the question-answer pair set $\Omega^{\mathcal{D}_3^m}$ corresponding to the sub-domain $\mathcal{D}_3^m$ in $\mathcal{P}^m$

        $\mathcal{B} = \mathcal{B} \cup \{(q^m, a^m)\}$

    Train the target model $\mathcal{M}^T$ using $\{(q^m, a^m)\}$

# C  Experimental Details

**Training Details** We introduce the specific experimental settings in the main experiments. For FuseChat, FuseChat-3.0, and Condor, we refer to the settings in their original papers [6, 7, 27], including the learning rate and training epochs. For SFT, we use a training schedule of 3 epochs and a consistent learning rate of $5e-6$ across all target models. Since Condor does not provide a learning rate adjustment scheme in its original paper, we uniformly set a cosine learning rate decay strategy for all baseline methods except FuseChat and FuseChat-3.0, with a warmup rate of $0.03$. For Bohdi, we also consistently use a constant learning rate of $5e-6$ across all target models. Given the limited number of samples per training round, we train the target model for 1 epoch based on the sampled $M$ question-answer pairs in each round. Additionally, apart from FuseChat providing training code, since FuseChat-3.0 and Condor do not provide training code, we uniformly use trl[3] as the training framework and train with bfloat16 precision. All of our training is conducted on $8\times$Nvidia A100 GPUs.

**Testing Details** To ensure unified and fair evaluation, we use opencompass[4] as the evaluation suite. In the generation settings for inference, we uniformly set a max input token length of 2048 for all methods. For MATH, GSM8K, HumanEval, and MBPP, which typically require longer reasoning outputs to solve, we uniformly use a max new token length of 4096. For MMLU, GPQA, TheoremQA, and BBH, which require shorter responses, we uniformly use a max new token length of 1024. To further mitigate the impact of randomness, we use greedy sampling for generation in all tests.

**Generation Settings** Regarding the temperature parameter settings during Bohdi's generation process, for models whose generation configs explicitly specify a temperature value—including Llama3.2-3B-Instruct and Llama3.1-8B-Instruct set to 0.6, Qwen2.5-7B-Instruct and Qwen2.5-14B-Instruct set to 0.7, and Mistral-Small-24B-Instruct-2501 set to 0.15—we follow the temperature parameters specified in their configs. For other models whose generation configs do not specify a temperature, including phi-4 and Gemma2-9B-IT, we uniformly apply a temperature of 0.7. During the domain proposal process in the Sprout operation and the question generation process in the Harvest operation, we use a top-p value of 0.95 across all models to encourage diversity in the generated domains and questions. In the answer generation process of the Harvest operation, we use a top-p value of 0.8 for all models to ensure response stability. Additionally, during the answer selection stage in Harvest, we apply a top-p value of 0.95 for the Leader model.

# D  Additional Experiments

**Fusion experiments on more target models** To further verify the universality of Bohdi, we conduct fusion experiments on two additional target models from different model classes and with different parameter sizes, Llama3.1-8B-Instruct [16] and the more powerful Qwen2.5-7B-Instruct [1]. We then compare them with other baselines. As shown in Tab. 3, the experimental results demonstrate that Bohdi continues to deliver outstanding performance on both target models. Notably for Llama3.1-8B-Instruct, Bohdi delivers uniform performance gains across all benchmarks. This advantage becomes particularly pronounced on GSM8K and TheoremQA, where all other baseline methods degrade model performance, Bohdi alone achieves positive enhancements. The most striking improvement occurs on TheoremQA, where Bohdi achieves a $3.76\%$ absolute performance gain over the base model. When Qwen2.5-7B-Instruct serves as the target model, due to its already sufficiently strong foundational capabilities, almost all methods lead to performance degradation in most benchmarks after fusion. Bohdi, however, still manages to bring about significant improvements in the target model's performance on MBPP and TheoremQA, without causing any noticeable decline on individual benchmarks. Furthermore, while other baselines exhibit average performance decay across each benchmarks, Bohdi continues to deliver an overall performance boost for Qwen2.5-7B-Instruct, highlighting its stability.

**The impact of the number of source models on domain allocation preferences** In Fig. 7 (a), (b), and (c), we show the domain hierarchy generated and corresponding sampling proportions during the Bohdi fusion process when using $K=1$ source model (Qwen2.5-14B-Instruct), $K=2$ source models (Qwen2.5-14B-Instruct and Mistral-Small-24B-Instruct-2501), and all three source

---

[3]https://github.com/huggingface/trl.git
[4]https://github.com/open-compass/opencompass.git

Table 3: Comparative results for Llama3.1-8B-Instruct [16] and Qwen2.5-7B-Instruct [1] as target models (accuracy in %). The best results and the second-best results are highlighted in bold and underlined, respectively. The numbers after the arrows indicate the absolute improvement (shown in red) or decline (shown in green) compared with the original model.

| Model | Data Volume | Multidisciplinary | | Mathematic | | Programming | | Reasoning | | AVG |
|---|---|---|---|---|---|---|---|---|---|---|
| | | MMLU (5-Shot) | GPQA (0-Shot CoT) | GSM8K (0-Shot CoT) | MATH (0-Shot CoT) | HumanEval (0-Shot) | MBPP (3-Shot) | TheoremQA (5-Shot) | BBH (3-Shot CoT) | |
| **Source Models** | | | | | | | | | | |
| Q-14B | – | 78.54 | 45.45 | 90.22 | 75.04 | 78.66 | 70.60 | 22.88 | 80.45 | 67.73 |
| M-24B | – | 80.95 | 46.46 | 91.89 | 68.84 | 79.88 | 68.20 | 37.12 | 82.71 | 69.51 |
| P-14B | – | 81.26 | 52.53 | 86.88 | 74.66 | 84.15 | 71.20 | 37.38 | 82.09 | 71.27 |
| **Target Model: Llama3.1-8B-Instruct** | | | | | | | | | | |
| Base | – | 67.01 | 29.29 | 79.61 | 45.88 | 65.85 | 55.20 | 31.75 | 66.47 | 55.13 |
| FuseChat | 95K | 67.31↑0.30 | 28.28↓1.01 | 79.23↓0.38 | 46.98↑1.10 | 67.68↑1.85 | 55.60↑0.40 | 31.25↓0.50 | 67.94↑1.47 | 55.53↑0.40 |
| SFT | 90K | 67.72↑0.71 | 33.84↑4.55 | 74.20↓5.41 | 50.08↑4.20 | 67.07↑1.22 | 56.40↑1.20 | 11.12↓20.63 | 56.25↓10.22 | 52.09↓3.04 |
| FuseChat-3.0 | 90K | 68.73↑1.72 | 33.33↑4.04 | 78.17↓1.44 | 53.46↑7.58 | 67.07↑1.22 | 55.60↑0.40 | 14.75↓17.00 | 59.98↓6.49 | 53.89↓1.24 |
| Condor | 90K | 68.65↑1.64 | 29.80↑0.51 | 75.88↓3.73 | 51.76↑5.88 | 65.85↑0.00 | 55.80↑0.60 | 14.37↓17.38 | 53.07↓13.40 | 51.90↓3.23 |
| Bohdi (Ours) | 1473 | 68.16↑1.15 | 32.32↑3.03 | 84.23↑4.62 | 52.58↑6.70 | 70.73↑4.88 | 58.80↑3.60 | 35.00↑3.25 | 69.31↑2.84 | 58.89↑3.76 |
| **Target Model: Qwen2.5-7B-Instruct** | | | | | | | | | | |
| Base | – | 73.34 | 37.88 | 87.11 | 69.68 | 82.32 | 63.00 | 19.38 | 72.37 | 63.14 |
| FuseChat | 95K | 73.52↑0.18 | 37.88↑0.00 | 87.41↑0.30 | 69.66↓0.02 | 77.44↓4.88 | 61.20↓1.80 | 19.88↑0.50 | 73.05↑0.68 | 62.51↓0.63 |
| SFT | 90K | 64.31↓9.03 | 34.34↓3.54 | 83.85↓3.26 | 65.46↓4.22 | 79.27↓3.05 | 65.40↑2.40 | 18.12↓1.26 | 69.21↓3.16 | 60.00↓3.14 |
| FuseChat-3.0 | 90K | 70.02↓3.32 | 34.85↓3.03 | 85.14↓1.97 | 68.96↓0.72 | 75.61↓6.71 | 62.20↓0.80 | 21.00↑1.62 | 71.14↓1.23 | 61.12↓2.02 |
| Condor | 90K | 61.86↓10.48 | 35.86↓2.02 | 83.15↓3.96 | 64.12↓5.56 | 71.95↓10.37 | 63.00↑0.00 | 12.38↓7.00 | 63.72↓8.65 | 57.01↓6.13 |
| Bohdi (Ours) | 1602 | 73.29↓0.05 | 35.35↓2.53 | 88.17↑1.06 | 70.28↑0.60 | 79.88↓2.44 | 66.80↑3.80 | 24.12↑4.74 | 71.83↓0.54 | 63.72↑0.58 |

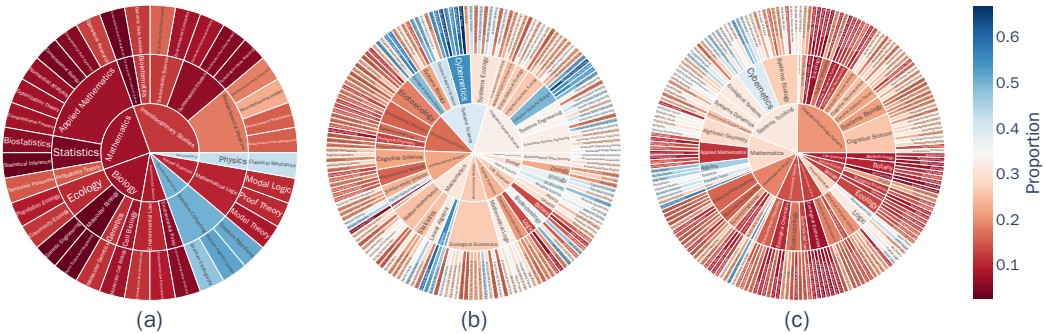

Figure 7: Impact of increasing the number of source models on domain allocation preferences. (a), (b), and (c) depict the domain hierarchy and corresponding sampling proportions generated by Bohdi during the fusion process for Llama3.2-3B-Instruct as the target model using $K = 1$ source model (Qwen2.5-14B-Instruct), $K = 2$ source models (Qwen2.5-14B-Instruct and Mistral-Small-24B-Instruct-2501), and all three source models, respectively.

models, with Llama3.2-3B-Instruct as the target model. As the results shown in Fig. 7, due to the inherent preferences of each model, fewer source models struggle to comprehensively evaluate the target model's capabilities across various dimensions. Consequently, with fewer source models, the domain sampling proportions exhibit a more pronounced bias. Especially when $K = 1$, the domain proportions are almost entirely concentrated on natural sciences and quantum mechanics, while other domains are consistently neglected. As the number of source models $K$ increases, the number of proposed domains also grows, and the proportioning of domain data becomes relatively more uniform, preventing any single domain from being excessively preferred.

**Bohdi's Fusion Efficiency** To further clarify Bohdi's fusion efficiency, we tally the runtime (in minutes) for each method across various models in the comparative experiments. All times are normalized to the runtime on a device setup of $8\times$Nvidia A100. For FuseChat, we include the preparatory time for fusion, encompassing the time consumed in representation extraction and vocabulary alignment stages, as these are integral parts of the fusion process. For SFT and FuseChat-3.0, we account for the time taken to collect responses from source models on the given dataset and the training time. For Condor, we consider the data generation time, including label synthesis, question and answer synthesis, and the answer refinement phase. For Bohdi, we include the time for all steps in its entire process, including domain proposal, question and answer generation, and answer selection by the leader model in the Meditation phase, as well as the training in the Enlightenment

Table 4: Runtime Comparison of Different Fusion Methods (in minutes)

| Method | FuseChat | SFT | FuseChat-3.0 | Condor | Bohdi (Ours) |
|---|---|---|---|---|---|
| Llama3.2-3B-Instruct | 2107 | 576 | 734 | 1433 | 515 |
| Llama3.1-8B-Instruct | 2581 | 674 | 927 | 1595 | 457 |
| Qwen2.5-7B-Instruct | 2392 | 642 | 856 | 1629 | 567 |
| Gemma2-9B-IT | 2917 | 773 | 1109 | 1651 | 547 |

phase. As shown in Tab. 4, Bohdi's execution time on each target model is significantly less than that of other methods, further highlighting its advantage in runtime efficiency. This indicates that Bohdi can more swiftly and conveniently bring about more pronounced improvements compared to other methods.

Table 5: Performance Comparison of Bohdi between Fixed Leader and Random Leader Selection (Target Model: Llama3.2-3B-Instruct).

| Strategy | MMLU | GPQA | GSM8K | MATH | HumanEval | MBPP | TheoremQA | BBH | AVG |
|---|---|---|---|---|---|---|---|---|---|
| Fix-Leader-Mistral | 60.43 | 35.35 | 75.51 | 45.08 | 56.10 | 45.0 | 24.38 | 51.3 | 49.14 |
| Fix-Leader-phi | 61.25 | 32.32 | 74.68 | 45.42 | 56.10 | 46.6 | 25.50 | 46.0 | 48.48 |
| Fix-Leader-Qwen | 61.01 | 23.23 | 75.06 | 46.52 | 57.93 | 48.8 | 25.87 | 54.9 | 49.17 |
| Random-Leader | **61.33** | **30.30** | **78.62** | **47.92** | **58.54** | **49.6** | **25.12** | **54.9** | **50.79** |

**The Impact of Random Leader Selection on the Stability of Model Fusion** To evaluate the impact of randomly selecting the leader model on fusion quality, we conduct comparative experiments using Llama3.2-3B-Instruct as the target model. Specifically, we successively fix each of the three source models—Mistral-Small-24B-Instruct-2501, phi-4, and Qwen2.5-14B-Instruct—as the sole leader and then compare these configurations against a strategy that randomly chooses the leader for every fusion step. The results shown in Tab 5 indicates that the random-leader strategy achieves the highest average score across the eight benchmarks ($50.79\%$) and exhibits noticeably more stable performance. Because any single leader tends to carry intrinsic judgment biases that can skew the fusion outcome, introducing randomness at the leader selection process helps cancel out these negative effects.

Table 6: Performance comparison (mean ± standard deviation) across three independent runs under aligned data budgets (1,782 samples).

| Method | Data | MMLU | GPQA | GSM8K | MATH | HumanEval | MBPP | TheoremQA | BBH | AVG |
|---|---|---|---|---|---|---|---|---|---|---|
| Base | — | 59.77 | 27.78 | 68.92 | 34.82 | 56.10 | 48.20 | 23.88 | 53.86 | 46.67 |
| FuseChat | 1782 | 59.84±0.06 | 27.44±1.05 | 69.22±0.15 | 35.52±0.32 | 54.68±1.27 | 47.40±0.40 | 24.21±0.39 | 54.17±0.19 | 46.56±0.16 |
| SFT | 1782 | 61.28±0.20 | 26.93±2.10 | 75.18±1.23 | 47.64±1.17 | 57.12±0.35 | 47.80±0.40 | 23.37±0.78 | 38.04±1.36 | 47.17±0.57 |
| FuseChat-3.0 | 1782 | 61.10±0.13 | 25.08±4.69 | 75.31±1.29 | 49.31±1.06 | 58.74±1.41 | 48.73±0.64 | 22.54±1.56 | 35.99±1.57 | 47.10±0.50 |
| Condor | 1782 | 60.70±0.07 | 27.79±2.16 | 71.45±0.87 | 46.58±0.38 | 57.32±0.61 | 48.33±0.58 | 25.17±0.50 | 30.60±0.99 | 45.99±0.44 |
| **Bohdi** | **1781.66±45.83** | **61.19±0.13** | **29.63±1.62** | **76.67±1.27** | **47.08±1.07** | **58.95±0.35** | **49.87±0.64** | **25.41±0.73** | **55.46±0.76** | **50.53±0.36** |

**Controlled Data Efficiency Comparison under Aligned Data Volumes** To ensure a fair comparison of different approaches under identical data budgets and to assess their stability, we conduct three independent runs of our proposed method, Bohdi, and unify the data usage of all baselines to 1782 samples—the rounded average of the data generated across Bohdi's three runs (1781.66). This guarantees that all methods are evaluated under approximately the same data budget. For FuseChat, we randomly sample 1782 instances from its generated dataset in each run; for the other baselines, we randomly select 594 samples from each of the three domains—mathematics, programming, and general—resulting in a total training set of 1782 samples. Under this aligned setting, we report the average performance and standard deviation across the three runs. Experimental results shown in Tab 6 that Bohdi achieves an average absolute performance gain of $3.86\%$ while maintaining low performance variance, significantly outperforming all baseline methods and demonstrating excellent data efficiency and stability. Moreover, thanks to its automated data ratio allocation mechanism, Bohdi exhibits smaller performance variance compared to other implicit fusion methods, indicating stronger robustness. It is worth noting that although the explicit fusion method FuseChat shows relatively small inter-run variance, its conservative fusion strategy fails to deliver noticeable performance improvements. These results fully validate the advantages of Bohdi in efficiently utilizing generated data and stably improving model performance.

# E Limitations

While the Bohdi framework has already demonstrated advantages in multiple aspects, including performance and efficiency, as a brand-new prototype framework, it still has the following two areas that could be further explored and improved: 1) It uses fixed instructions for data generation throughout the fusion process, which to some extent limits the diversity of the generated data. Considering the introduction of adaptive prompt design schemes could further promote more comprehensive and diverse data generation and knowledge injection. 2) Further improvement strategies such as self-refinement or multi-agent debate could be considered to help enhance data quality and more deeply extract knowledge from the source models. We will further explore and attempt the above two points in our subsequent research work.

# F Related Instruction Formats

Note: In the formats below, the **bold text** enclosed in "{}" represents input variables.

## F.1 $p\left[\textbf{Expand}, \textbf{Parent}\left(\mathcal{D}_{i,j}\right)\right]$ in The Sprout Operation

---

**For Main Domain Generation**

I need to generate a hierarchical systematic knowledge tree. First, I need to determine a set of main subject domains, please use your world knowledge to propose a **Main Domain** that systematically taught in primary/secondary/higher education (e.g., in exact sciences, computer engineering, or other natural sciences and humanities), which should be as broad as possible to cover a wide range of child domains.
**STRICT REQUIREMENTS**:
1. Must propose **EXACTLY ONE** new domain name
2. The proposed domain must be a clearly defined academic field related to **natural sciences** (such as physics, chemistry), **social sciences** (such as law, philosophy), **humanities** (linguistics, art), **formal sciences** (such as mathematics, computer science), or **interdisciplinary** fields (such as medicine, social psychology, etc.).
**STRICT RESPONSE FORMAT**: The proposed domain must be enclosed between [Proposition Start] and [Proposition End], following the format below:
[Proposition Start]proposed domain[Proposition End]
Now, please provide your proposed domain according to the requirements mentioned above.

---

**For Secondary Domain Generation**

This is a path of a hierarchical systematic knowledge tree: {**Sampled Main Domain**}, and now you need to propose a subject domain that logically and structurally follows this path, i.e., the domain you propose must be a secondary domain of {**Sampled Main Domain**}.
**STRICT REQUIREMENTS**:
1. Must propose **EXACTLY ONE** new domain name
2. The proposed domain must be a clearly defined academic field related to **natural sciences** (such as physics, chemistry), **social sciences** (such as law, philosophy), **humanities** (linguistics, art), **formal sciences** (such as mathematics, computer science), or **interdisciplinary** fields (such as medicine, social psychology, etc.).
**STRICT RESPONSE FORMAT**: The proposed domain must be enclosed between [Proposition Start] and [Proposition End], following the format below:
[Proposition Start]proposed domain[Proposition End]
Now, please provide your proposed domain according to the requirements mentioned above.

---

**For Sub-Domain Generation**

This is a path of a hierarchical systematic knowledge tree: {**Sampled Main Domain**} → {**Sampled Secondary Domain**}, and now you need to propose a subject domain that logically and structurally follows this path, i.e., the domain you propose must be a specific sub-domain of {**Sampled Secondary Domain**}.
**STRICT REQUIREMENTS**:
1. Must propose **EXACTLY ONE** new domain name
2. The proposed domain must be a clearly defined academic field related to **natural sciences** (such as physics, chemistry), **social sciences** (such as law, philosophy), **humanities** (linguistics, art), **formal sciences** (such as mathematics, computer science), or **interdisciplinary** fields (such as medicine, social psychology, etc.).
**STRICT RESPONSE FORMAT**: The proposed domain must be enclosed between [Proposition Start] and [Proposition End], following the format below:
[Proposition Start]proposed domain[Proposition End]
Now, please provide your proposed domain according to the requirements mentioned above.

## F.2 $p$ [**Inquiry**; $\mathcal{P}$] and $p$[**Select**; $q$, $\mathcal{A}^q$] in The Harvest Operation

---

**$p$ [Inquiry; $\mathcal{P}$] for Question Generation**

Now I need to create high-quality SFT data for LLM training, so I need you to generate such data.

For now, **you only need to create one question**. I will provide you with a specified main domain, its secondary domain, and a further refined sub-domain in the format [Main Domain]→[Secondary domain]→[Sub-Domain].

The corresponding topic is:

{**Sampled Main Domain**} → {**Sampled Secondary Domain**} → {**Sampled Sub-Domain**}

The question must meet these requirements:
1. Strictly fall within the scope of [Sub-Domain] - neither too broad nor too narrow, and the stem of the question should first contain sufficient background information or relevant conditions
2. The question you provide should be a relatively challenging, but it must be solvable, and the answer should be definitive
3. {**Selected Style**}
4. Must be as original and concise as possible
5. The expression style of the question should be **as diverse as possible**
6. Enclose your response strictly between [Question Start] and [Question End] as shown below:

[Question Start]Question[Question End]

Now provide **EXACTLY ONE** question for the sub-domain **{Sampled Sub-Domain}** within secondary domain **{Sampled Secondary Domain}** of main domain **{Sampled Main Domain}**.

---

Note: The aforementioned {**Selected Style**} is randomly drawn from the following question type style prompts in each question generation process:

---

**Candidate Question Style Prompts**

- The question should be a high-difficulty one that requires a step-by-step solution, with the answer numbered accordingly.
- The question should be open-ended and require the answer to include at least two different perspectives.
- The question should require coding to solve, with the answer presented in Markdown code block format.
- The question should require comparative analysis, with the answer displayed in a table format to show pros and cons.
- The question should require association with knowledge from other fields (e.g., math + music).
- The question should be styled as casual conversation and Q&A in daily life, with the tone and speaking style of the reply specified (e.g., using metaphors, rhyming).

---

## $p[\textbf{Select}; q, \mathcal{A}^q]$ for Response Selection

Please compare and evaluate the quality of the multiple answers to the following question, and return the index of the best one using **Arabic numerals**:

**Question**

{**Generated Quesition**}

**Answers to be Evaluated**

{**Candidate Answers**}

**Evaluation Criteria**
1. Accuracy (40%): Whether the answer correctly solves the problem and match the required response style.
2. Clarity (30%): The expression is structured clearly and smoothly, and the content is concise without being verbose.
3. Completeness (20%): Whether all necessary details are covered.
4. Relevance (10%): Whether the answer closely relates to the question.

**Additional Constraints**
Incoherent and repetitive answers should be directly assigned a low score, regardless of whether they contain the correct answer.

**Output Requirements**
1. You only need to reply with which answer you consider to be the best answer, and return the index (1-digit) of the best answer enclosed between [Best Answer Start] and [Best Answer End], following the format below:

The best answer is [Best Answer Start]index[Best Answer End].

2. Only one index for the best answer can be returned.

# G Examples of Bohdi's Synthetic Data

Here we present the synthesized data generated during the Bohdi fusion process from our main experiments. We label the title of each colorbox containing an example with its corresponding branch (Main Domain → Secondary Domain → Sub Domain).

---

**Mathematics → Algebra → Linear Algebra**

**Question**:

```
Given two matrices \\( A = \\begin{pmatrix} 1 & 2 \\\\ 3 & 4 \\end{pmatrix}
\\) and \\( B = \\begin{pmatrix} -1 & 1 \\\\ 0 & 2 \\end{pmatrix} \\),
compare their eigenvalues and eigenvectors. Discuss the implications for
diagonalizability and invertibility in a table format.
```

**Answer**:

```
To compare the eigenvalues and eigenvectors of the given matrices \\( A \\)
and \\( B \\), and to discuss their implications for diagonalizability and
invertibility, we first need to calculate the eigenvalues and eigenvectors of
 each matrix.\n\n### Matrix \\( A \\)\nGiven \\( A = \\begin{pmatrix} 1 & 2
\\\\ 3 & 4 \\end{pmatrix} \\).\n\n#### Eigenvalues of \\( A \\)\nThe
characteristic equation for matrix \\( A \\) is given by \\( \\det(A - \\
lambda I) = 0 \\):\n\\[\n\\det\\begin{pmatrix} 1-\\lambda & 2 \\\\ 3 & 4-\\
lambda \\end{pmatrix} = (1-\\lambda)(4-\\lambda) - 6 = \\lambda^2 - 5\\lambda
 - 2 = 0\n\\]\nSolving the quadratic equation, we get:\n\\[\n\\lambda = \\
frac{5 \\pm \\sqrt{25 + 8}}{2} = \\frac{5 \\pm \\sqrt{33}}{2}\n\\]\nSo, the
eigenvalues are \\( \\lambda_1 = \\frac{5 + \\sqrt{33}}{2} \\) and \\( \\
lambda_2 = \\frac{5 - \\sqrt{33}}{2} \\).\n#### Eigenvectors of \\( A \\)\
nFor \\( \\lambda_1 \\):\n\\[\n\\begin{pmatrix} 1-\\lambda_1 & 2 \\\\ 3 &
4-\\lambda_1 \\end{pmatrix} \\begin{pmatrix} x \\\\ y \\end{pmatrix} = \\
begin{pmatrix} 0 \\\\ 0 \\end{pmatrix}\n\\]\nSolving this system, we get an
eigenvector corresponding to \\( \\lambda_1 \\).\nFor \\( \\lambda_2 \\):\n
\\[\n\\begin{pmatrix} 1-\\lambda_2 & 2 \\\\ 3 & 4-\\lambda_2 \\end{pmatrix}
\\begin{pmatrix} x \\\\ y \\end{pmatrix} = \\begin{pmatrix} 0 \\\\ 0 \\end{
pmatrix}\n\\]\nSolving this system, we get an eigenvector corresponding to
\\( \\lambda_2 \\).\n\nSince \\( A \\) has two distinct eigenvalues, it is
diagonalizable.\n\n#### Invertibility of \\( A \\)\nSince \\( A \\) is a \\(
2 \\times 2 \\) matrix with two distinct eigenvalues, neither of which is
zero, \\( A \\) is invertible.\n\n### Matrix \\( B \\)\nGiven \\( B = \\begin
{pmatrix} -1 & 1 \\\\ 0 & 2 \\end{pmatrix} \\).\n\n#### Eigenvalues of \\( B
\\)\nThe characteristic equation for matrix \\( B \\) is given by \\( \\det(B
 - \\lambda I) = 0 \\):\n\\[\n\\det\\begin{pmatrix} -1-\\lambda & 1 \\\\ 0 &
2-\\lambda \\end{pmatrix} = (-1-\\lambda)(2-\\lambda) = 0\n\\]\nSo, the
eigenvalues are \\( \\lambda_1 = -1 \\) and \\( \\lambda_2 = 2 \\).\n\n####
Eigenvectors of \\( B \\)\nFor \\( \\lambda_1 = -1 \\):\n\\[\n\\begin{pmatrix
} -1-(-1) & 1 \\\\ 0 & 2-(-1) \\end{pmatrix} \\begin{pmatrix} x \\\\ y \\end{
pmatrix} = \\begin{pmatrix} 0 & 1 \\\\ 0 & 3 \\end{pmatrix} \\begin{pmatrix}
x \\\\ y \\end{pmatrix} = \\begin{pmatrix} 0 \\\\ 0 \\end{pmatrix}\n\\]\
nSolving this system, we get an eigenvector corresponding to \\( \\lambda_1 =
 -1 \\).\n\nFor \\( \\lambda_2 = 2 \\):\n\\[\n\\begin{pmatrix} -1-2 & 1 \\\\
0 & 2-2 \\end{pmatrix} \\begin{pmatrix} x \\\\ y \\end{pmatrix} = \\begin{
pmatrix} -3 & 1 \\\\ 0 & 0 \\end{pmatrix} \\begin{pmatrix} x \\\\ y \\end{
pmatrix} = \\begin{pmatrix} 0 \\\\ 0 \\end{pmatrix}\n\\]\nSolving this system,
 we get an eigenvector corresponding to \\( \\lambda_2 = 2 \\).\n\nSince \\(
B \\) has two distinct eigenvalues, it is diagonalizable.\n\n####
Invertibility of \\( B \\)\nSince \\( B \\) is a \\( 2 \\times 2 \\) matrix
with two distinct eigenvalues, neither of which is zero, \\( B \\) is
invertible.\n\n### Summary Table\n| Matrix | Eigenvalues | Diagonalizable |
Invertible |\n|--------|-------------|----------------|------------|\n| \\( A
 \\) | \\( \\frac{5 + \\sqrt{33}}{2}, \\frac{5 - \\sqrt{33}}{2} \\) | Yes |
Yes |\n| \\( B \\) | \\( -1, 2 \\) | Yes | Yes |\n\n
```

**Biology → Genetics → Molecular Genetics**

**Question**:

Given a DNA sequence represented as a string, write a Python function to find the positions of all occurrences of a specific palindromic DNA sequence (e.g., \"CGGCG\") within the main sequence. The function should return a list of tuples, where each tuple contains the start and end indices of each occurrence. For instance, if the input DNA sequence is \"ACCGGCGTACCGGCGT\", and the target palindromic sequence is \"CGGCG\", the function should return [(2, 6), (9, 13)]. Note: In a DNA sequence, the indices are 0-based, and the sequence is read from 5' to 3'.

**Answer**:

```
def find_palindromic_sequences(sequence, pattern):\n \"\"\"\n Finds all
occurrences of a specific palindromic DNA sequence within a given DNA
sequence.\n\n Args:\n sequence (str): The DNA sequence to search in.\n
pattern (str): The palindromic sequence to search for.\n\n Returns:\n list[
tuple[int, int]]: A list of tuples, where each tuple contains the start and
end indices of each occurrence.\n\n \"\"\"\n # Initialize an empty list to
store the positions of the occurrences\n positions = []\n\n # Iterate over
the sequence with a sliding window of the pattern's length\n for i in range(
len(sequence) - len(pattern) + 1):\n # Check if the current window matches
the pattern\n if sequence[i:i + len(pattern)] == pattern:\n # If it matches,
add the start and end indices to the positions list\n positions.append((i, i
+ len(pattern)))\n\n # Return the list of positions\n return positions\n\n\n#
 Example usage:\nsequence = \"ACCGGCGTACCGGCGT\"\npattern = \"CGGCG\"\nprint(
find_palindromic_sequences(sequence, pattern)) # Output: [(2, 6), (9, 13)]
```

**Formal Sciences → Theoretical Computer Science → Computational Complexity Theory**

**Question:**

Consider a computational problem where you are given a Boolean formula in Conjunctive Normal Form (CNF) with n variables and m clauses. From the perspective of computational complexity theory, discuss how the difficulty of determining whether there exists an assignment of truth values to these variables that makes the formula true, varies as a function of both n and m. Additionally, explore how the problem's complexity might differ if the clauses are restricted to having exactly three literals each, forming a 3-SAT problem.

**Answer:**

The problem of determining whether there exists an assignment of truth values to variables that makes a Boolean formula in Conjunctive Normal Form (CNF) true is known as the Satisfiability problem (SAT). SAT is a fundamental problem in computational complexity theory and is the first problem to be proven NP-complete.\n\n### General CNF SAT\nThe complexity of the general SAT problem as a function of the number of variables \\( n \\) and the number of clauses \\( m \\) is not straightforward. The problem is NP-complete, meaning that any instance of the problem can be solved in polynomial time relative to the size of the input if and only if P = NP, which is a major unsolved problem in computer science. In practice, the difficulty of solving SAT instances can vary greatly depending on the structure of the CNF formula. Generally, as \\( n \\) and \\( m \\) increase, the problem becomes harder to solve in the worst case. However, the exact relationship between the difficulty and the parameters \\( n \\) and \\( m \\) is not well understood, and there are instances of SAT problems that are hard even for small \\( n \\) and large \\( m \\), and vice versa.\n\n### 3-SAT\nWhen the clauses in the CNF formula are restricted to have exactly three literals each, the problem is known as 3-SAT. This is a special case of SAT and is also NP-complete. The restriction to three literals per clause does not make the problem easier from a complexity-theoretic perspective; 3-SAT remains NP-complete. However, practical algorithms for solving 3-SAT can sometimes be more efficient than for general SAT because of the restriction on clause size. The problem's structure can sometimes be exploited by specific algorithms to reduce the search space.\n\n### Complexity as a Function of \\( n \\) and \\( m \\)\n- **General SAT**: The complexity of SAT increases with both \\( n \\) and \\( m \\), but the exact relationship is not well-defined. The problem is in NP, and no polynomial-time algorithm is known for solving it in the general case.\n- **3-SAT**: Similarly, the complexity of 3-SAT increases with both \\( n \\) and \\( m \\). However, due to the restriction on clause size, certain algorithms may perform better on average or for certain types of instances.\n\n### Practical Implications\nIn practice, the difficulty of solving SAT and 3-SAT instances can be highly dependent on the specific structure of the problem instance. For example, instances with many clauses compared to variables might be easier to solve than those with fewer clauses. Similarly, instances with certain types of variable interactions might be easier or harder to solve. Advanced algorithms like Conflict-Driven Clause Learning (CDCL) SAT solvers exploit such structures to solve large SAT instances efficiently in practice.\n\n

Environmental Science → Environmental Law → Environmental Litigation

**Question**:

Consider a hypothetical scenario where a company is facing a lawsuit for violating environmental regulations. The lawsuit alleges that the company emitted a total of 200 tons of pollutants over a period of 5 years, exceeding the permissible limit by 50 tons. The company claims that due to a software error, the emissions data was incorrectly recorded, and the actual emissions were only 150 tons. Assuming the permissible limit is 175 tons per 5 years, and the company needs to prove its claim using statistical evidence. Write a Python script to simulate the emissions data for 5 years, considering a normal distribution of daily emissions with a mean of 10 tons and a standard deviation of 2 tons, and calculate the total emissions to validate the company's claim.

**Answer**:

This script should also include a function to calculate the probability that the total emissions exceed the permissible limit.
```python
import numpy as np

# Function to simulate emissions data
def simulate_emissions(years, mean, std_dev):
    days_per_year = 365
    total_days = years * days_per_year
    emissions = np.random.normal(mean, std_dev, total_days)
    return emissions

# Function to calculate total emissions
def calculate_total_emissions(emissions):
    return np.sum(emissions)

# Function to calculate the probability of exceeding the permissible limit
def calculate_exceedance_probability(emissions, permissible_limit):
    total_emissions = calculate_total_emissions(emissions)
    exceedance_probability = np.mean(total_emissions > permissible_limit)
    return exceedance_probability

# Parameters
years = 5
mean_emissions = 10
std_dev_emissions = 2
permissible_limit = 175

# Simulate emissions data
emissions_data = simulate_emissions(years, mean_emissions, std_dev_emissions)

# Calculate total emissions
total_emissions = calculate_total_emissions(emissions_data)

# Calculate the probability of exceeding the permissible limit
exceedance_probability = calculate_exceedance_probability(emissions_data, permissible_limit)

# Print results
print(f"Total Emissions: {total_emissions} tons")
print(f"Probability of Exceeding Permissible Limit: {exceedance_probability:.4f}")
```

## Natural Sciences → Physics → Quantum Mechanics

**Question**:

Imagine you are tasked with simulating a two-level quantum system, often referred to as a qubit, in a magnetic field. The Hamiltonian of such a system can be described as $$H = \frac{\hbar \omega}{2} \sigma_z$$, where $$\hbar$$ is the reduced Planck constant, $$\omega$$ is the angular frequency, and $$\sigma_z$$ is the Pauli z-matrix. Given an initial state $$\left|\psi(0)\right> = \frac{1}{\sqrt{2}}(\left|0\right> + \left|1\right>)$$, find the state $$\left|\psi(t)\right>$$ at time $$t$$. Write a Python function that calculates the state at any given time $$t$$ using the provided parameters. Assume $$\hbar = 1$$ for simplicity.

**Answer**:

To solve the problem of finding the state of a qubit in a magnetic field at any given time $$t$$, we can use the time-evolution operator $$U(t) = e^{-iHt/\hbar}$$. Given the Hamiltonian $$H = \frac{\hbar \omega}{2} \sigma_z$$ and setting $$\hbar = 1$$, we get $$H = \frac{\omega}{2} \sigma_z$$.

The Pauli z-matrix $$\sigma_z$$ is:
$$\sigma_z = \begin{pmatrix}1 & 0 \\ 0 & -1\end{pmatrix}$$
Thus, the Hamiltonian becomes:
$$H = \frac{\omega}{2} \begin{pmatrix}1 & 0 \\ 0 & -1\end{pmatrix}$$

The time-evolution operator $$U(t)$$ can be written as:
$$U(t) = e^{-iHt} = e^{-i(\frac{\omega}{2} \sigma_z)t} = e^{-i\frac{\omega t}{2} \sigma_z}$$

For a diagonal matrix like $$\sigma_z$$, the exponential of the matrix is simply the exponential of each diagonal element:
$$e^{-i\frac{\omega t}{2} \sigma_z} = \begin{pmatrix}e^{-i\frac{\omega t}{2}} & 0 \\ 0 & e^{i\frac{\omega t}{2}}\end{pmatrix}$$

Given the initial state $$\left|\psi(0)\right> = \frac{1}{\sqrt{2}}(\left|0\right> + \left|1\right>)$$, we can write it as a column vector:
$$\left|\psi(0)\right> = \frac{1}{\sqrt{2}} \begin{pmatrix}1 \\ 1\end{pmatrix}$$

To find the state at time $$t$$, we apply the time-evolution operator $$U(t)$$ to the initial state:
$$\left|\psi(t)\right> = U(t) \left|\psi(0)\right>$$

Let's write the Python function to calculate this:

```python
import numpy as np

def calculate_state(t, omega):
    """
    Calculates the state of the qubit at time t given the angular frequency omega.

    Parameters:
    t (float): Time at which to calculate the state.
    omega (float): Angular frequency of the magnetic field.

    Returns:
    np.ndarray: The state vector at time t.
    """
    # Define the time-evolution operator
    U_t = np.array([[np.exp(-1j * omega * t / 2), 0],
                    [0, np.exp(1j * omega * t / 2)]])

    # Initial state vector
    psi_0 = np.array([1, 1]) / np.sqrt(2)

    # Calculate the state at time t
    psi_t = np.dot(U_t, psi_0)

    return psi_t

# Example usage:
t = 1.0 # Time
omega = 1.0 # Angular frequency
psi_t = calculate_state(t, omega)
print("State at time t:", psi_t)
```

