# OpenReview forum: "Bohdi: Heterogeneous LLM Fusion with Automatic Data Exploration"
_NeurIPS.cc/2025/Conference — NeurIPS 2025 poster_

### Official Review · Reviewer_wt9t · 2025-07-02

**Clarity:** 3
**Significance:** 3
**Originality:** 3
**Rating:** 4
**Confidence:** 4

**Summary:**

The paper introduces the Bohdi framework for heterogeneous LLM fusion, which at its core is a HMAB framework for domain exploration and adaptive data collection. To further improve on the original HMAB framework, it adds the Introspection-Rebirth mechanism, which is essentially to re-estimate the target's model's performance utilizing a binomial likelihood ratio test. The overall framework is an iterative two-stage process: (1). HMAB for parameter update; (2). sampling and target model training. Experiments show Bohdi significantly outperforms baselines (different EF, IF fusion methods), demonstrating much better data efficiency (~1.x%)  and reduced capability imbalance among various tasks.

**Questions:**

- Most of the questions are in the Weakness Section.

- One question re the baseline methods, for each question, instead of collecting "golden responses" from all three source models, can we use a leader / reward model to select one response and train on it? What's the performance there? It could give a clearer picture on the benefit of using HMAB.

- The paper mentions two main motivations, i.e., drawback from existing LLM fusion methods, one is fixed data selection, this is resolved by HMAB; the other is limited domain coverage, could you make it clearer how the Bohdi framework address this in an effective way?

**Ethical Concerns:**

["NO or VERY MINOR ethics concerns only"]

**Final Justification:**

I will keep my score unchanged, the score is already positive.

**Quality:**

3

**Strengths And Weaknesses:**

Strength:

- The paper is well-motivated, outlined the two main drawbacks of the popular LLM fusion methods: (1). limited domain coverage; and (2). fixed data allocation; this makes bandit framework a naturally choice to update the posterior of the target model's performance and perform adaptive sampling.

- The results are pretty strong, it shows better performance compared with existing data fusion baselines (i.e., FuseChat, FuseChat-3.0, Condor), while at the same time has much better sample efficiency, with only utilizing ~1.x% of the data in training of the target LLM.

- The paper is well-written, the method is explained in a very clear manner, the HMAB algorithm, sprout and harvest operations, IR mechanism with sliding window, etc, the figure also helps the illustration.


Weakness:

- Some of the details are missing, for example, how to construct the knowledge graph tree in detail, what's the granularity there? More details regarding the reproducibility can be added, such as the number of nodes in each level, how are they determined, etc. Also, how do we determine if a question falls into which category? Could you give more details on the "unk" arm?

- While the improvement in sample efficiency is impressive, the performance in each benchmark seems lack statistical significance. It would be great to add error bars on some of the tasks, at least, if experimental costs is a concern.

- The results on Bohdi with an external reward model seems perform worse, and it somehow shows that Bohdi is pretty sensitive to the quality of the reward model, it might be a concern of the method. It would be great if the authors could either add more ablations regards to the robustness respect to the reward model, or arguing why choosing the leader model is always good here.

- It would be great if the paper could mention the computational / inference cost by running the HMAB algorithm, by making a fair comparison with other baseline methods.

---

> ### Author Rebuttal · Authors · 2025-07-31
>
> # Response To Reviewer wt9t
> We greatly appreciate the insightful comments and suggestions from the reviewer. Below, we will provide point-by-point responses to each of your questions and recommendations.
>
> ### Response To Weakness #1
>
> We appreciate the reviewer's insightful feedback.
>
> **Knowledge Tree Granularity and Node Organization:** The knowledge tree is constructed with three layers. It originates from a `[Root]` node, branching into distinct Main, Secondary, and Sub-level domains. You can visualize its specific structure in Figure 7 of our appendix within the supplementary materials.
>
> **New Node Creation:** During each meditation phase, starting from the root, our Thompson Sampling (TS) algorithm samples child nodes from existing ones. When a sampled child node is `unk` (e.g., sampling `unk` from the `Mathematics` node), it signifies that the corresponding parent node (`Mathematics`) requires new sub-node expansion. At this point, a **Sprout operation** is executed, prompting the source model to propose new domains to be added as sub-domains under `Mathematics`. For more specific operational details, please refer to the algorithm pseudocode in Appendix B.4.
>
> **Node Category Determination:** The resulting node tree is stored as a dictionary, which clearly defines the hierarchical relationships. During the meditation phase, after a path of nodes (Main → Secondary → Sub) is sampled, we perform a Harvest operation in parallel for each path to generate data corresponding to that path. For instance, if data A is sampled along the path `Mathematics` (Root) → `Algebra` (Secondary) → `Linear Algebra` (Sub), it will be stored in the list corresponding to the innermost key (`Linear Algebra`) of that path in the dictionary. This allows for clear querying of its associated Main, Secondary, and and Sub domains.
>
> **The `unk` Arm:** Simply put, the `unk` arm represents the need for "current parent node expansion." Each node, upon creation, is automatically assigned an initialized `unk` arm, and these `unk` arms are independent across nodes. When the `unk` arm is sampled from node 'a', a Sprout operation is executed to generate new child nodes for node 'a'. You can find clear details regarding the specific execution of this operation in the algorithm pseudocode in Appendix B.4.
>
> ### Response To Weakness #2
> We deeply value the reviewer's insightful comments. Through a series of comprehensive experiments, we have compellingly demonstrated the consistency and stability of Bohdi's performance.
>
> **In the comparative experiments**, we conducted a thorough evaluation of all methods across four target models with different architectures and parameter sizes: Llama3.2-3B-Instruct, Qwen2.5-7B-Instruct, Gemma2-9B-IT, and Llama3.1-8B-Instruct. As shown in Table 1 of the main text and Table 2 in the appendix, Bohdi consistently outperforms other baseline methods across different target models.
>
> **In the ablation experiments**, we further explored Bohdi's performance under different source model configurations. When using more parameter-efficient source models (Llama3.2-3B-Instruct, Qwen2.5-3B-Instruct, and Gemma2-2B-IT) and fusing them with the relatively larger Gemma2-9B-IT as the target model, the results still indicate that the target model fused by Bohdi can effectively absorb the advantages of the source models. This is particularly evident in tasks where Gemma2-9B-IT originally performed weaker than the source models, such as GPQA and TheoremQA.
>
> **Regarding the organization of generated domains**, as visually illustrated in Appendix Figure 7, when Llama3.2-3B-Instruct serves as the target model, the domains generated by Bohdi consistently include the vast majority of domains from scenarios with fewer source models as the number of source models increases. This indicates that Bohdi maintains stability in terms of domain expansion.
>
> Furthermore, throughout the evaluation process, we uniformly adopted greedy sampling to minimize the impact of autoregressive generation randomness on the evaluation results, ensuring the rigor of the assessment, as elaborated in Appendix C.
>
> The **consistent improvements observed across these diverse evaluations and experimental results fully validate Bohdi's remarkable stability under various conditions**. In light of your valuable feedback, we will incorporate these analyses and findings into the main text as much as possible in the final version, enabling readers to more intuitively grasp these significant improvements. We deeply appreciate your valuable input.
>
> ### Response To Weakness #3
> We deeply value the reviewer's valuable feedback. **Bohdi is designed to evaluate feedback by randomly selecting a Leader Model from among the source models, rather than relying on an external reward model**. The primary motivation for including an ablation study with an external reward model is to validate the effectiveness of the random Leader Model selection scheme. Following the reviewer's suggestion, we conducted an additional set of experiments comparing the current approach of randomly selecting a Leader Model with fixing each of the three source models as the Leader Model, using Llama3.2-3B-Instruct as the target model. **The results, presented in the table below, demonstrate that random selection of the Leader Model yields more consistent average performance improvements**. This is because each model inherently possesses judgment biases that could lead to performance imbalances, and random Leader selection better mitigates the negative impacts of such biases. We have incorporated this experiment into our revised version. Your valuable input is crucial to enhancing the quality of our work.
>
> | Strategy              | MMLU  | GPQA | GSM8K | MATH | HumanEval | MBPP | TheoremQA | BBH  | AVG       |
> |-----------------------|-------|------|-------|------|-----------|------|-----------|------|-----------|
> | Fix-Leader-Mixtral    | 60.43 | 35.35| 75.51 | 45.08| 56.1      | 45   | 24.38     | 51.3 | 49.14  |
> | Fix-Leader-phi        | 61.25 | 32.32| 74.68 | 45.42| 56.1      | 46.6 | 25.5      | 46   | 48.48  |
> | Fix-Leader-Qwen       | 61.01 | 23.23| 75.06 | 46.52| 57.93     | 48.8 | 25.87     | 54.9 | 49.17    |
> | Random-Leader         | 61.33 | 30.3 | 78.62 | 47.92| 58.54     | 49.6 | 25.12     | 54.9 | 50.79  |
>
>
> ### Response To Weakness #4
> We deeply value the reviewer's valuable suggestions. In Table 3 of the appendix to our paper, we present statistics on the actual total runtime (in minutes) of each baseline method and Bohdi on each target model when using 8 A100 GPUs, allowing for a fair comparison of the practical costs of each method. Across all target models, Bohdi's actual runtime consistently remains shorter than that of other baselines. This result effectively demonstrates Bohdi's practical efficiency advantages over other baselines. In the final version of the paper, we will also move this result into the main text to more clearly present Bohdi's efficiency advantages to readers. Your valuable suggestions are greatly appreciated.
>
>
> ### Response To Question #1
> We deeply value the reviewer‘s insightful questions.
>
> **Bohdi's response generation takes place within the Harvest operation**. In fact, **we adopt the approach you described: the Leader Model (including the reward model in the ablation study) is used to select the superior answer from the responses of the three source models, rather than leveraging all source model responses collectively for training**.
>
> **The role of HMAB here is not targeted at the problem itself, but constructed at the granularity of problem domains**. When sampling training data, a batch of paths is sampled via the TS algorithm, and data generated during the meditation phase under each corresponding path is drawn to construct the training set. Further details of this process can be found in the description of the Sprout and Harvest operations in Appendix B, as well as the pseudocode in Appendix B4.
>
> ### Response To Question #2
> We deeply value the reviewer's valuable suggestions.
>
> Regarding **the issue of limited domain coverage**: This constraint is addressed through the self-construction of the knowledge tree in HMAB. By enabling autonomous construction of a structured knowledge tree among models, comprehensive and systematic organization of problem domains is achieved automatically, ensuring the breadth of domain knowledge. Meanwhile, the Harvest operation allows for data synthesis tailored to each domain path, thereby resolving the issue of missing open-source data in fine-grained domains.
>
> Regarding **the problem of fixed data allocation**: This limitation is overcome through dynamic adjustment of the posterior distribution of reward distribution parameters for each arm in HMAB. Based on the performance evaluation of the target model and source models across different domains, the alpha and beta parameters of these arms are dynamically adjusted. This facilitates a shift in data allocation toward domains where the target model performs weaker relative to the source models, enabling more rational distribution.
>
> We will elaborate on these two points in greater detail in the main text of the final version. Your valuable suggestions are crucial for enhancing the clarity of our work.

---

> > ### Author Response · Authors · 2025-08-04
> >
> > Dear Reviewer,
> >
> > We sincerely appreciate the time and effort you have dedicated to reviewing our manuscript. In response to your valuable comments and suggestions, we have provided point-by-point replies to each of your queries.
> >
> > As the discussion period will conclude in two days, we kindly request that you review our responses at your convenience. Should any additional questions or suggestions arise, please let us know and we will address them immediately. Once again, we are grateful for your feedback, which plays a crucial role in enhancing the quality of our work.
> >
> > Sincerely,
> >
> > The Authors of Manuscript #28761

---

> > ### Comment · Reviewer_wt9t · 2025-08-08
> > **Response to the authors**
> >
> > Thank you to the authors for the detailed and thoughtful rebuttal. I appreciate the effort to address my questions and the new experiments conducted, which have clarified several aspects of the work.

---

> > > ### Author Response · Authors · 2025-08-09
> > >
> > > Dear Reviewer wt9t,
> > >
> > > We appreciate your patient review and response! If you have any other questions or suggestions, please let us know, and we will address them promptly. We again appreciate the effort you have put into reviewing our manuscript!
> > >
> > > Best Wishes,
> > >
> > > Authors of Manuscript #28761

---

### Official Review · Reviewer_3FuQ · 2025-07-03

**Clarity:** 2
**Significance:** 2
**Originality:** 2
**Rating:** 4
**Confidence:** 4

**Summary:**

The paper introduces Bohdi, a framework that fuses knowledge from several heterogeneous source LLMs into a smaller target LLM without using any real data. Bohdi (1) arranges knowledge domains in a three-level tree, (2) explores new sub-domains through a Sprout operation and generates synthetic Q-A pairs through a Harvest operation, and (3) frames domain selection and data-allocation as a Hierarchical Multi-Armed Bandit solved by a DynaBranches variant of Thompson Sampling, equipped with an Introspection-Rebirth mechanism that resets posteriors via a sliding-window BLRT when the target model’s ability shifts.

**Questions:**

How many total tokens were generated by Sprout + Harvest over 50 rounds? Please provide an approximate compute cost (GPU-hours).

Did you check for overlap between harvested Q-A pairs and any of the evaluation test sets (particularly GSM8K and MATH)?

**Ethical Concerns:**

["NO or VERY MINOR ethics concerns only"]

**Final Justification:**

While I'm willing to raise my score to 4, I will not actively defend the paper if other reviewers or the AC express concerns.

**Limitations:**

Please refer to the Weaknesses and Questions.

**Quality:**

2

**Strengths And Weaknesses:**

Pros:

1. Casting domain expansion + data-ratio selection as an HMAB and augmenting Thompson Sampling with IR is interesting and new in the LLM-fusion context.

2. The authors evaluate their method on diverse benchmarks across knowledge, coding, math and reasoning; includes ablations on IR window/quantile, data split ratios.

3. The authors use two orders of magnitude fewer training samples than competing fusion or SFT baselines yet often matches or beats them.

Cons:

1. The improvements over SOTA methods are incremental. Also since only single runs are reported and the authors explicitly acknowledge the absence of error bars due to cost. Therefore the variance in evaluations can be large.

2. Condor and FuseChat-3.0 rely on explicit reward models or preference optimization. The authors disable reward models in Bohdi’s main results, yet the strongest baselines in recent literature (e.g., InfiFusion, SuperMario) are not included or are lightly tuned.

3. While Bohdi uses only 1.7K target-training samples, it repeatedly queries three ≥ 14B source models for Sprout/Harvest. The total token budget and wall-clock cost are not reported.

---

> ### Author Rebuttal · Authors · 2025-07-31
>
> # Response To Reviewer 3FuQ
> We appreciate the reviewer's valuable comments and suggestions. We will provide point-by-point responses to each of your questions below.
>
> ### Response To Cons #1
> We appreciate the reviewer's valuable suggestions.
>
> Regarding **the performance improvements of our method**, it is evident from the experiments on all four target models (results in Table 1 of the main text and Table 2 of the appendix) that Bohdi achieves the optimal performance improvement on average. Moreover, it accomplishes this with significantly less data than other baselines and without relying on any real external data, which is no easy feat under such conditions.
>
> Even setting these prerequisites aside, the performance improvements brought by Bohdi are still prominent. For instance, compared to explicit fusion baselines like FuseChat, Bohdi delivers remarkably significant performance gains in single domains. For example, on TheoremQA, Bohdi brings a 72.09% relative improvement to the target model Gemma2-9B-IT compared to the original model, while FuseChat only achieves a 6.40% relative improvement. When compared to other implicit fusion baselines such as FuseChat3.0, Bohdi's advantages remain evident, mainly reflected in more balanced performance improvements across tasks and better average performance. For example, when the target model is Llama3.2-3B-Instruct, the model fused with Bohdi shows performance gains in every domain, and in most cases, achieves the optimal improvement—particularly a 9.07% relative improvement on GPQA. In contrast, FuseChat3.0 exhibits significant performance declines on GPQA, TheoremQA, and BBH, with a 20.01% decline specifically on GPQA.
>
> Thus, **the performance improvements brought by our method are relatively significant, especially considering the prerequisites that it does not rely on any real data and uses far more data-efficiently than baseline methods**.
>
> Regarding **the stability of the evaluation**, we have strongly demonstrated the consistency and stability of Bohdi's performance through multiple comprehensive experiments. For example, in the comparative experiment dimension, we compared all methods on four target models with different structures and parameter sizes: Llama3.2-3B-Instruct, Qwen2.5-7B-Instruct, Gemma2-9B-IT, and Llama3.1-8B-Instruct. The results reported in Table 1 of the main text and Table 2 of the appendix show that Bohdi stably outperforms other baselines across different target models. In the ablation experiments, we also tried using several more compact source models (Llama3.2-3B-Instruct, Qwen2.5-3B-Instruct, and Gemma2-2B-IT) and conducted fusion experiments with the larger Gemma2-9B-IT as the target model. The results indicate that the target model fused with Bohdi can still effectively absorb the advantages of the source models, particularly evident in tasks where Gemma2-9B-IT itself performs weakly, such as GPQA and TheoremQA. Additionally, in Figure 7 of the appendix, we visualized the generated domain structure organization when Llama3.2-3B is used as the target model. As the number of source models increases, the generated domains still include most of the domains observed when there are fewer source models, which strongly illustrates the stability of Bohdi in domain expansion. **The consistent improvements across these diverse evaluations and experimental results fully demonstrate Bohdi's stability under different conditions**. Furthermore, during the evaluation, we used greedy sampling to minimize the impact of randomness in autoregressive generation, as described in Appendix C. Following your suggestion, in the final version, **we will incorporate these analyses and results into the main text as much as possible to help readers understand these improvements more intuitively**.
>
> We again appreciate your valuable suggestions, which have been very helpful in enhancing the clarity of our paper.
>
> ### Response To Cons #2
> We appreciate the reviewer’s insightful comments.
>
> Regarding Condor and FuseChat3.0, only FuseChat3.0 relies on a reward model, whereas Condor does not [1]. In our comparison, **we retained FuseChat3.0’s reward model exactly as prescribed in the original paper, as stated in our experimental setup**. Bohdi itself is designed to operate without any additional reward model.
>
> Concerning **the two baselines you mentioned**, we are unsure if the SuperMario you referred to refers to the DARE method in [2]. **This is a homogeneous model fusion scheme, which only applies to scenarios where the models involved in fusion have identical structures and parameter sizes**. It is **not applicable to the heterogeneous model fusion setting we are researching**.
>
> As for InfiFusion [3], no code has been made publicly available in its published paper, making it impossible for us to reproduce it. Additionally, as an explicit LLM fusion scheme, the performance reported in the InfiFusion paper does not show significant improvements compared to FuseChat, the baseline we selected, and its design is similar to that of FuseChat. Therefore, the results of FuseChat can, to some extent, roughly represent the performance of InfiFusion. We have further clarified this point in the revised version of the paper. We again appreciate your valuable suggestions.
>
> [1]Condor: Enhance LLM Alignment with Knowledge-Driven Data Synthesis and Refinement, in ACL 25.
>
> [2]Language models are super mario: Absorbing abilities from homologous models as a free lunch, in ICML 24.
>
> [3]InfiFusion: A Unified Framework for Enhanced Cross-Model Reasoning via LLM Fusion
>
> ### Response To Cons #3 & Question #1
> We appreciate the reviewer's insightful feedback. Among the comparative baselines, explicit model fusion baselines do not actually generate text during operation, making it impossible to directly compare the number of generated tokens across methods. However, in Table 3 of the appendix, we have provided statistics on the actual total running time (in minutes) of each baseline method and Bohdi on each target model when using 8 A100 GPUs, allowing for a fair comparison of the practical efficiency of each method. **Across all target models, Bohdi's actual running time is consistently shorter than that of other baselines**. This result practically demonstrates Bohdi's efficiency advantage over other baselines. In the final version of the paper, we will move this result to the main text to present Bohdi's efficiency advantage more clearly to readers. We again appreciate the reviewer's valuable suggestions, which are very important for enhancing the quality of our work.
>
> ### Response To Question #2
> We appreciate the reviewer's valuable question. In response to your inquiry, we compared the data generated by Bohdi with the test task data across all four target model settings, and found that the number of overlaps between the generated data and the test data is zero. This further confirms that **there is no overlap between the collected QA pairs and the test data**. We have further elaborated on this result in the experimental section of the revised manuscript. We again appreciate the reviewer's insightful question.

---

> > ### Author Response · Authors · 2025-08-04
> >
> > Dear Reviewer,
> >
> > We sincerely appreciate the time and effort you have dedicated to reviewing our manuscript. In response to your valuable comments and suggestions, we have provided point-by-point replies to each of your queries.
> >
> > As the discussion period will conclude in two days, we kindly request that you review our responses at your convenience. Should any additional questions or suggestions arise, please let us know and we will address them immediately. Once again, we are grateful for your feedback, which plays a crucial role in enhancing the quality of our work.
> >
> > Sincerely,
> >
> > The Authors of Manuscript #28761

---

> > > ### Comment · Reviewer_3FuQ · 2025-08-07
> > >
> > > Q1 concern remain unaddressed. The claimed significance of Bohdi's improvements is unsupported for two primary reasons:
> > >
> > > 1. Questionable Significance and Unproven Scaling: The absolute gains are marginal, and presenting them as relative percentages is misleading. The key claim of "data efficiency" is unsubstantiated without a direct comparison to a baseline trained on the same amount of data, leaving the method's scalability unproven.
> > >
> > > 2. Lack of Statistical Rigor: All results are from single runs, making it impossible to assess statistical significance. Given the high variance in LLM benchmarks, the small reported gains are indistinguishable from random noise.
> > >
> > > In short, the combination of marginal results, unproven scaling, and a lack of statistical validation means the paper fails to demonstrate a robust or meaningful advantage for the Bohdi method. I appreciate the emphasis on data efficiency, but a comparison using the same number of data points is missing, which would help demonstrate whether Bohdi can scale.

---

> > > > ### Author Response · Authors · 2025-08-08
> > > >
> > > > Dear Reviewer 3FuQ,
> > > >
> > > > We truly appreciate your thorough review and detailed feedback on our response, and we also appreciate your more specific and detailed suggestions! We will address each of them below.
> > > >
> > > > 1) **Regarding the significance of absolute gain**: In terms of absolute gain, compared with other heterogeneous LLM fusion methods, the increment achieved by Bohdi is also relatively significant. Bohdi can bring an absolute average performance improvement of over 4% to the original model. For reference, in other heterogeneous LLM fusion works, [1] reports an absolute average performance improvement of approximately 1-2% compared to the target model, [2] achieves a 1.72% absolute average performance improvement, and [3] obtains an average absolute performance improvement of 3-7% when using over 100K real data and guided by a much larger 72B source model. Therefore, in comparison, **the absolute gain of Bohdi is indeed significant**. Once again, we truly appreciate the reviewer's valuable feedback.
> > > >
> > > > 2) **Rationale for originally using relative gain**: We initially chose the relative gain based on the evaluation practices observed in relevant works like [1], [2], and [4]. Following your advice, **we have now replaced all labeled gains with absolute gains throughout the revised manuscript**. We are confident this modification will enable readers to more precisely understand the actual improvements achieved by our method. Your insightful recommendation has been most helpful in enhancing our work.
> > > >
> > > > 3) **Regarding comparison under the same data volume and error bar from multiple runs**: We appreciate the reviewer's valuable suggestions. Following your advice, we ran Bohdi three times on the target model Llama3.2-3B-Instruct, and set the data volume for other baselines to 1782—the rounded average of the data volumes generated from Bohdi's three runs (1781.66 in the table below)—to ensure nearly identical data volume settings. For FuseChat, we randomly sampled 1782 pieces of data from its dataset each time; for other baselines, we randomly selected 594 samples (1782/3) from each of the three domains (mathematics, programming, and general domains) to form the 1782 samples of data used for training. **Under this aligned setting, we calculated the average results across the three runs along with their standard deviations, as shown in the table below. Bohdi demonstrates relatively stable average gains, delivering a 3.86% absolute improvement that significantly outperforms other baselines, still exhibits an excellent data efficiency**. Furthermore, thanks to the automatic data proportioning, the average performance fluctuation of Bohdi is better compared to other implicit fusion baselines. While the explicit fusion baseline FuseChat has a smaller standard deviation, its overly conservative updates result in no significant gains. These results have been incorporated into our revised manuscript. We are particularly grateful for your insightful suggestions that have significantly strengthened our empirical validation.
> > > >
> > > > | | Data Volume | MMLU  | GPQA  | GSM8K | MATH  | HumanEval | MBPP| TheoremQA | BBH | AVG        |
> > > > |-|--|-|--|--|--|--|--|---|--|--|
> > > > | Base        | -        | 59.77    | 27.78    | 68.92    | 34.82     | 56.1     | 48.2      | 23.88    | 53.86 | 46.67 |
> > > > | FuseChat    | 1782 | 59.84±0.06 | 27.44±1.05 | 69.22±0.15 | 35.52±0.32 | 54.68±1.27 | 47.40±0.4 | 24.21±0.39 | 54.17±0.19 | 46.56±0.16 |
> > > > | SFT         | 1782 | 61.28±0.20 | 26.93±2.10 | 75.18±1.23 | 47.64±1.17 | 57.12±0.35 | 47.80±0.4 | 23.37±0.78 | 38.04±1.36 | 47.17±0.57 |
> > > > | FuseChat-3.0 | 1782 | 61.10±0.13 | 25.08±4.69 | 75.31±1.29 | 49.31±1.06 | 58.74±1.41 | 48.73±0.64 | 22.54±1.56 | 35.99±1.57 | 47.10±0.50 |
> > > > | Condor      | 1782 | 60.70±0.07 | 27.79±2.16 | 71.45±0.87 | 46.58±0.38 | 57.32±0.61 | 48.33±0.58 | 25.17±0.50 | 30.6±0.99 | 45.99±0.44 |
> > > > | Bohdi       | 1781.66±45.83 | 61.19±0.13 | 29.63±1.62 | 76.67±1.27 | 47.08±1.07 | 58.95±0.35 | 49.87±0.64 | 25.41±0.73 | 55.46±0.76 | 50.53±0.36 |
> > > >
> > > > Finally, we once again appreciate your valuable suggestions, which have greatly helped improve the quality of our work. We also wish to express our gratitude for the energy and time you have invested in patient communication with us! If you have any other suggestions or questions, we will make every effort to address them promptly.
> > > >
> > > > [1]KNOWLEDGE FUSION OF LARGE LANGUAGE MODELS.
> > > >
> > > > [2]Probabilistic Token Alignment for Large Language Model Fusion.
> > > >
> > > > [3]FUSECHAT-3.0: PREFERENCE OPTIMIZATION MEETS HETEROGENEOUS MODEL FUSION.
> > > >
> > > > [4]FUSECHAT: Knowledge Fusion of Chat Models.

---

> > > > > ### Comment · Reviewer_3FuQ · 2025-08-08
> > > > >
> > > > > Thank you for your thoughtful and timely response. Your clarifications have addressed most of my concerns about experimental results. I have no further questions and will adjust my score accordingly.

---

> > > > > > ### Author Response · Authors · 2025-08-08
> > > > > >
> > > > > > Dear Reviewer 3FuQ,
> > > > > >
> > > > > > We are delighted to hear that we have addressed your concerns, and we feel truly honored! Once again, we deeply appreciate your valuable suggestions and comments, as well as the energy and time you have invested in engaging in the discussion.
> > > > > >
> > > > > > Best Wishes,
> > > > > >
> > > > > > The Authors of Manuscript #28761

---

### Official Review · Reviewer_ShRp · 2025-07-04

**Clarity:** 3
**Significance:** 2
**Originality:** 4
**Rating:** 4
**Confidence:** 4

**Summary:**

This paper proposes Bohdi, a heterogeneous large language model fusion framework based only on synthetic data. Bohdi systematically organizes domain knowledge into a three-level structured knowledge tree. It formulates domain expansion and data ratio allocation as a hierarchical multi-armed bandit (HMAB) problem and solves it using the proposed DynaBranches mechanism. Bohdi contains an introspection-rebirth (IR) module that adaptively adjusts domain sampling using a sliding window binomial likelihood ratio test (SWBLRT) based on performance feedback.

**Questions:**

- Can you rovide detailed output and evaluation of intermediate steps? I would like to know the numerical details and detailed examples of the new data created in the paper. Rather than just seeing mathematical proofs, I want to see actual proof that this method can be implemented.

- The paper mentions 5 different models, but only evaluates two of them as target models, and there is randomness in the selection of the method, which raises questions about the stability and generalizability of the method.
- What are the memory, computation, and latency costs as the number of domains and source LLMs increase? Is there a practical limit to the granularity or number of domains that Bohdi can handle?

**Ethical Concerns:**

["NO or VERY MINOR ethics concerns only"]

**Final Justification:**

After carefully checking out the authors' response, I think my initial score still reflects my evaluation after checking out the author's rebuttal, so I am keeping it unchanged.

**Limitations:**

yes

**Quality:**

3

**Strengths And Weaknesses:**

**Strengths:**

- It is claimed that Bohdi is the first framework exclusively using synthetic data for heterogeneous LLM fusion and data can be allocated dynamically among different LLMs.
- The method in this article is novel and clearly explained, as well as using clear mathematical symbols.
- The amount of experimental data is large, especially the ablation study, the angles are diverse, and there are experimental verifications in different fields and different models.

**Weaknesses:**
- One of the innovations of the entire framework is to synthesize data without using existing data, but the article rarely discusses or conducts experimental analysis on the quality and correctness of the synthesized data, as well as comprehensiveness and reproducibility of newly generated data. In addition, this method of synthesizing data has a lot of randomness, which is mainly reflected in the selection of the Leader model in Harvest and the selection of the sub-field with the highest sampling value according to the Beta distribution in DynaBranches as the next direction to be explored. The article does not provide a detailed analysis of these intermediate results and randomness.
- The source models used in the experiments vary significantly in parameter scale, and the target models come from different companies and also differ in size. This raises concerns about the consistency and fairness of the experimental setup. If the goal is to demonstrate that your method is adaptable across different models, the experiments should first be conducted on models of the same size. Only one variable should be changed at a time in order to ensure a valid controlled comparison. And then you can report the results containing variable settings.
- The construction and management of a hierarchical tree, the need to run multiple source models for every “Sprout” and “Harvest” operation, and the use of adaptive sampling across potentially hundreds of domains (as illustrated in Figure 6) could become significant bottlenecks when scaling to very large source LLM pools or fine-grained domains. However, potential challenges related to compute, memory, or latency are not addressed in the main text.

---

> ### Author Rebuttal · Authors · 2025-07-31
>
> # Response To Reviewer ShRp
> We appreciate the reviewer's meticulous examination and valuable feedback, which are of great significance to our manuscript. Below, we will provide point-by-point responses to each of your comments and suggestions.
>
> ### Response To Weakness #1
> We appreciate the reviewer's valuable comments. In line with your feedback, we have added a dedicated section in the appendix of the revised manuscript to present and discuss the quality of synthetic data. Here, we briefly showcase a piece of data from the path Mathematics - Algebra - Linear Algebra. Due to space constraints, we have omitted part of the response here:
>
> Example:
> Q: Consider two matrices, $A = \\begin{pmatrix} 2 & 4 \\\\ 3 & 6 \\end{pmatrix}$ and $B = \\begin{pmatrix} 1 & 2 \\\\ 0.5 & 1 \\end{pmatrix}$. Compare the eigenvalues and eigenvectors of $A$ and $B$. Discuss the properties of these matrices based on their eigenvalues and eigenvectors, and summarize your findings in a table format.
>
> A: ...... Matrix A: ...... **Eigenvalues**: \n   The characteristic equation for matrix $A$ is given by ...... Thus, the eigenvalues are $\\lambda_1 = 0$ and $\\lambda_2 = 8$.\n\n2. **Eigenvectors**:\n   - For $\\lambda_1 = 0$:\n     ...... we can choose $x = 2$ and $y = -1$ as a solution ...... ### Matrix B: ......\n\n### Summary Table\n\n| Property/Matrix | A                     | B                     |\n|---|---|---|\n| Eigenvalues     | 0, 8                  | 0, 2                  |\n| Eigenvectors    | (2, -1), (2, 3)       | (2, -1), (2, 1)       |\n| Properties      | $A$ has one zero eigenvalue and one non-zero eigenvalue. It is singular and non-invertible.<br> $A$'s eigenvectors are not orthogonal. ...... |\n\n
>
> The question in the above example data involves comparing the eigenvalues of a matrix and requires an answer in table format. The generated response correctly solves this problem and strictly follows the requirement to present the answer in a table, thus demonstrating full correctness and completeness, which indicates that the synthetic data is of high quality.
>
> Furthermore, the randomness introduced by the Beta distribution sampling serves as the foundation for ensuring diversified and parallel domain exploration, as discussed in Section 2.3. However, it does not unduly affect the stability of the algorithm. Existing research has also provided solid guarantees for the convergence of such sampling algorithms [1][2]. Meanwhile, in the ablation experiment illustrated in Figure 7 of the supplementary materials, we visualized the domain structure generated when Llama3.2-3B-Instruct is used as the target model, with statistics showing that as the number of source models increases, the generated domains still encompass most of the domains observed when there are fewer source models. This strongly confirms the stability of the overall domain expansion.
>
> Regarding the impact of randomly selecting the Leader Model, we conducted experiments where each of the three source models was fixed as the Leader Model, and compared these with the current scheme of randomly selecting the Leader Model, using Llama3.2-3B-Instruct as the target model. The results presented in the table below show that randomly selecting the Leader Model yields more stable performance. Since each model has inherent judgment biases, which can lead to unbalanced performance, random selection of the Leader can better counteract the negative effects of such biases. Therefore, the randomness introduced by the random selection of the Leader Model and the Beta distribution random sampling does not have a negative impact on the stability of the algorithm.
>
> | Strategy              | MMLU  | GPQA | GSM8K | MATH | HumanEval | MBPP | TheoremQA | BBH  | AVG       |
> |----|--|--|--|------|---|-|---|------|---|
> | Fix-Leader-Mistral    | 60.43 | 35.35| 75.51 | 45.08| 56.1      | 45   | 24.38     | 51.3 | 49.14  |
> | Fix-Leader-phi        | 61.25 | 32.32| 74.68 | 45.42| 56.1      | 46.6 | 25.5      | 46   | 48.48  |
> | Fix-Leader-Qwen       | 61.01 | 23.23| 75.06 | 46.52| 57.93     | 48.8 | 25.87     | 54.9 | 49.17    |
> | Random-Leader         | 61.33 | 30.3 | 78.62 | 47.92| 58.54     | 49.6 | 25.12     | 54.9 | 50.79  |
>
> [1]Analysis of Thompson Sampling for the Multi-armed Bandit Problem, in JMLR.
>
> [2]An Information-Theoretic Analysis for Thompson Sampling with Many Actions, in NeurIPS.
>
> ### Response To Weakness #2
> We appreciate the reviewer's constructive suggestions, which are crucial for enhancing the quality of our work.
>
> We appreciate the reviewer's constructive suggestions. The primary purpose of the structural differences between the source models and target model used is to ensure heterogeneity between models, which aligns with the focus of our research.
>
> Regarding the **selection logic for target models**, we aimed to cover the most widely accessed and used mainstream model types, thus choosing Qwen2.5-7B-Instruct, Gemma2-9B-IT, and Llama3.1-8B-Instruct. These three models, with comparable parameter sizes, form the main body of our experiments. To further verify the effectiveness and generalization ability of our method, we also included Llama3.2-3B-Instruct, a model with a smaller parameter size, as a target model. Due to space constraints, these results are reported in Table 1 of the main text and Table 2 of the appendix, fully demonstrating the universality of our method across target models of different structures and scales.
>
> In summary, we have **fully considered controlled comparisons of parameter sizes in model selection**, including both model combinations with similar parameter sizes and models of different scales. Meanwhile, selecting models from **different companies** is intended to **ensure the premise of heterogeneity** and strengthen the persuasiveness of our experimental results by using the most mainstream open-source models available. Furthermore, in **each set of experiments, all compared methods were conducted under the same source and target model settings**, thus ensuring the fairness and consistency of the experiments. We have clearly elaborated on the above content in the revised version. We again appreciate your valuable suggestions, which are vital for improving the clarity of our work.
>
> ### Response To Weakness #3 & Question #3
> We appreciate the reviewer's insightful comments. Regarding the cost of constructing and managing the hierarchical tree, we will address this from two dimensions: the number of models and the number of domains.
>
> 1) **Concerning the number of models**: When the number of involved LLMs increases, the cost rises linearly with the number of LLMs. However, any scheme built on multiple LLMs will inevitably face this objective cost increase due to a larger number of LLMs, which is also true for all the methods we compared with. New domains generated through the Sprout operation are fixed as new components of the tree after generation, so Sprout is typically executed a limited number of times. The response collection in the Harvest operation is only related to the amount of generated data—for each piece of data generated, responses from each model need to be collected once. Similar operations are equally necessary for other baselines, whether explicitly or implicitly fused. Thus, while an increase in model parameters leads to higher costs, this is an unavoidable objective cost increase for any current method.
>
> 2) **Concerning the cost from an increase in the number of domains**: The computational cost brought by more domains is mainly reflected in the sampling process of the TS algorithm. However, this process does not require direct participation of any LLM; it only involves variable sampling based on the Beta distribution parameters. The computational effort required here is merely O(1) floating-point operations per data point. Even when sampling across 1000 domains, this can be completed in less than 0.001 seconds on a regular consumer-grade CPU. Moreover, our method involves sampling across at most several hundred domains (see Figure 6 in the main text). Therefore, the efficiency cost from an increase in the number of domains and their management is almost negligible. Additionally, in Table 3 of our Appendix, we have provided statistics on the actual total running time (in minutes) of each baseline method and Bohdi on each target model when using 8 A100 GPUs, which practically demonstrates Bohdi's efficiency advantage over other baselines.
>
> We again appreciate your valuable comments.
>
> ### Response To Question #1
> We appreciate your valuable question. In response to this, all instructions involved in Bohdi can be found in Appendix F. For examples of data and analysis of data quality, please refer to our response to weakness #1.
>
> ### Response To Question #2
> We appreciate your insightful comments.
>
> Regarding the comprehensiveness of target models, we aimed to cover the most widely accessed and used mainstream model types, thus selecting four target models: Qwen2.5-7B-Instruct, Gemma2-9B-IT, Llama3.1-8B-Instruct, and Llama3.2-3B-Instruct. Due to space constraints, the results with Llama3.1-8B-Instruct and Qwen2.5-7B-Instruct as target models are recorded in Table 2 of the appendix, where Bohdi still shows significant advantages over other baselines. In the final version of the paper, we will also move this part of the results to the main text.
>
> As for the logic behind selecting baselines, we chose FuseChat3.0, the current state-of-the-art implicit heterogeneous model fusion baseline, and FuseChat, the most representative explicit fusion baseline. Additionally, to validate the effectiveness of Bohdi as an implicit model fusion scheme, we included a baseline that directly collects data for SFT and Condor, a baseline that performs data synthesis based on a static label tree. We have further emphasized this baseline selection logic in the experimental section of the revised version. We again appreciate your valuable suggestions.

---

> > ### Author Response · Authors · 2025-08-04
> >
> > Dear Reviewer,
> >
> > We sincerely appreciate the time and effort you have dedicated to reviewing our manuscript. In response to your valuable comments and suggestions, we have provided point-by-point replies to each of your queries.
> >
> > As the discussion period will conclude in two days, we kindly request that you review our responses at your convenience. Should any additional questions or suggestions arise, please let us know and we will address them immediately. Once again, we are grateful for your feedback, which plays a crucial role in enhancing the quality of our work.
> >
> > Sincerely,
> >
> > The Authors of Manuscript #28761

---

> > ### Comment · Reviewer_ShRp · 2025-08-08
> >
> > Thank you for the additional experimental results and clarifications! I think my initial score still reflects my evaluation after checking out the author's rebuttal, so I am keeping it unchanged.

---

> > > ### Author Response · Authors · 2025-08-09
> > >
> > > Dear Reviewer ShRp,
> > >
> > > We appreciate your patient review and response! If you have any other questions or suggestions, please let us know, and we will address them promptly. We again appreciate the effort you have put into reviewing our manuscript!
> > >
> > > Best Wishes,
> > >
> > > Authors of Manuscript #28761

---

### Official Review · Reviewer_YY3V · 2025-07-05

**Clarity:** 3
**Significance:** 4
**Originality:** 4
**Rating:** 5
**Confidence:** 3

**Summary:**

This paper presents a novel framework for fusing knowledge from heterogeneous LLMs into a compact target model without real data. It addresses key limitations of existing methods, including limited domain coverage and capability imbalance,  via organizing knowledge into a hierarchical tree and automating domain exploration/data generation using multi-model collaboration. The key innovations include dynamic domain expansion and data allocation as a Hierarchical Multi-Armed Bandit problem, solved by the DynaBranches mechanism with online capability tracking. Experiments demonstrate significant empirical improvements. Overall, this is a very good work. But without the available code, it is not easy to fully justify this framework.

**Questions:**

Check the weakness part mentioned above.

**Ethical Concerns:**

["NO or VERY MINOR ethics concerns only"]

**Final Justification:**

I am satified with the author's response. My final rating is accept.

**Limitations:**

Check the weakness part mentioned above.

**Quality:**

3

**Strengths And Weaknesses:**

Strength:
1. Removes dependence on any real or in-domain dataset, allowing the target LLM to inherit knowledge that is both broader and privacy-safe.
2. Adaptive allocation virtually eliminates capability imbalance and achieves markedly higher data efficiency than static-mix baselines.
3. Online adaptation yields consistent performance gains across diverse benchmarks, outperforming previous fusion methods on every evaluated target LLM.

Weakness:
1. The major concern about this work is that the framework is quite complex and without the available codes, it makes the justification and  and replication challenging.
2. The magnitudes of improvement and decline reported in Table 1 are unclear, as the reference baseline against which they are calculated is not specified.
3. The designations “H1” and “H2” in lines 140–142 do not align with the labels shown in the corresponding figure, which may cause confusion.
4. The influence of choosing the auxiliary model H2 at random has not been explored; a dedicated ablation study is needed to assess its impact.

---

> ### Author Rebuttal · Authors · 2025-07-31
>
> # Response To Reviewer YY3V
>
> We sincerely appreciate the reviewer’s thorough and exacting review. Your comments and suggestions are pivotal to improving our manuscript. We provide a point-by-point response below.
>
> ### Response To Weakness #1
> We appreciate the reviewer's insightful feedback.
>
> For comprehensive design, Bohdi incorporates multiple components and mechanisms. To ensure readers fully grasp its functionality, we've utilized various presentation methods. Beyond the main methodology in the body of the paper and operational details in the appendix of the supplementary materials, we include **schematic diagrams** for each component and the overall framework (Figures 2 and 3 in the main text). Crucially, Appendix B.4 provides a **highly detailed pseodocode of the entire algorithm's workflow**, and Appendix F contains **all associated LLM instructions**. We believe this offers readers a thorough understanding of our algorithm's operational logic and specific procedures.
>
> Regarding code availability, **we commit to including a link to the open-source code in the final version of the paper**. This will clearly outline all running, evaluation, and environment setup instructions, along with links to the **Bohdi-fused model weights**, as promised in our checklist. We hope you understand our decision not to directly include the full code at this stage, as it serves to protect our original work. We again appreciate your valuable comments.
>
> ### Response To Weakness #2
> We appreciate the reviewer's valuable feedback.
>
> To clarify, all our reported improvements and declines are determined **relative to the base model's performance on its corresponding task**. For instance, when the target model is Llama3.2-3B-Instruct, Bohdi improves its BBH accuracy to 54.90% from the vanilla Llama3.2-3B-Instruct's 53.86%, marking a 1.93% relative increase over the original model. For overall average performance, the Bohdi-fused Llama3.2-3B-Instruct achieves an average accuracy of 50.79%, which is an 8.83% relative improvement compared to the vanilla Llama3.2-3B-Instruct's 46.67%. We've explicitly noted this clarification in the revised manuscript to enhance reader understanding. We again appreciate the reviewer's valuable suggestions, which have been highly helpful in enhancing the clarity of our paper.
>
> ### Response To Weakness #3
> We appreciate the reviewer's meticulous examination and insightful suggestions. We have taken note of this point and made revisions accordingly in the updated version. Additionally, we have conducted a thorough review and polishing of the entire manuscript. We again appreciate your insightful comments, which play an important role in refining the details of our work.
>
> ### Response To Weakness #4
> We appreciate the reviewer's constructive suggestions. Following your advice, we conducted experiments where each of the three source models was fixed as the Leader Model, and compared these with the current scheme of randomly selecting the Leader Model, using Llama3.2-3B-Instruct as the target model. The results are presented in the table below. In comparison, the scheme of randomly selecting the Leader Model demonstrates more balanced performance gains and achieves the best average (AVG) performance. While fixing a single model as the Leader may yield better results in certain individual tasks than random selection, it often leads to unbalanced performance due to the inherent biases of each model. Randomly selecting the Leader can better counteract the negative impacts of such biases and alleviate the issue of performance imbalance.
>
> | Strategy              | MMLU  | GPQA | GSM8K | MATH | HumanEval | MBPP | TheoremQA | BBH  | AVG       |
> |-----------------------|-------|------|-------|------|-----------|------|-----------|------|-----------|
> | Fix-Leader-Mistral    | 60.43 | 35.35| 75.51 | 45.08| 56.1      | 45   | 24.38     | 51.3 | 49.14  |
> | Fix-Leader-phi        | 61.25 | 32.32| 74.68 | 45.42| 56.1      | 46.6 | 25.5      | 46   | 48.48  |
> | Fix-Leader-Qwen       | 61.01 | 23.23| 75.06 | 46.52| 57.93     | 48.8 | 25.87     | 54.9 | 49.17    |
> | Random-Leader         | 61.33 | 30.3 | 78.62 | 47.92| 58.54     | 49.6 | 25.12     | 54.9 | 50.79  |
>
> Furthermore, in the current version of the manuscript, we have also verified the approach of using a Reward Model for Answer evaluation in the ablation experiment shown in Figure 4. The results indicate that randomly selecting a Leader Model from the source models for evaluation remains a more optimal strategy. We again appreciate the reviewer's valuable suggestions, which are of great significance for enhancing the quality of our work.

---

> > ### Author Response · Authors · 2025-08-04
> >
> > Dear Reviewer,
> >
> > We sincerely appreciate the time and effort you have dedicated to reviewing our manuscript. In response to your valuable comments and suggestions, we have provided point-by-point replies to each of your queries.
> >
> > As the discussion period will conclude in two days, we kindly request that you review our responses at your convenience. Should any additional questions or suggestions arise, please let us know and we will address them immediately. Once again, we are grateful for your feedback, which plays a crucial role in enhancing the quality of our work.
> >
> > Sincerely,
> >
> > The Authors of Manuscript #28761

---

### Note · Authors · 2025-08-13

Dear AC and Reviewers,

We sincerely appreciate the thorough review and valuable feedback from all reviewers on our work. In particular, we are grateful for their recognition of **the novelty of our work** (Reviewers ShRp, 3FuQ), **the breakthroughs of our method in eliminating reliance on real-world data and achieving adaptive data allocation** (Reviewers YY3V, ShRp), **the clarity of our writing** (Reviewers ShRp and wt9t), **the consistency and high data efficiency of our results** (Reviewers YY3V, ShRp, wt9t, and 3FuQ), and **the comprehensiveness of our experimental evaluation** (Reviewer ShRp).

The reviewers' questions primarily focused on: 1) the statistical significance of the experimental results, 2) the practical computational cost of the method, and 3) the absence of available code.

Regarding 1), **we compared the performance of all baselines under aligned data volumes, reporting the mean and standard deviation of three experimental runs, our method still outperforms other baselines**. These results have been reported in our discussion with Reviewer 3FuQ.

Regarding 2), we provided the actual total runtime of each method in Table 3 of the submitted appendix, showing that **our method remains lower in running cost than others**. Regarding the posterior-sampling overhead inside the TS algorithm, **we clarified to reviewer ShRp that each draw takes on the order of $10^{-3}$ seconds and is therefore negligible**.

Regarding 3), **we have committed in our checklist and responses to reviewer YY3V that the final version of the paper will include links to the open-source code and the fused Bohdi model weights**.

**All new experiments have been incorporated into our revised manuscript, and we have also made modifications and refinements to the paper based on the reviewers' suggestions**.

As a heterogeneous LLM fusion study with well-justified motivations and novel methodological design, we are particularly encouraged by the reviewers' recognition of Bohdi. Their constructive suggestions have been instrumental in further enhancing the quality of our paper. We also believe our work holds promise for broader applications, such as continual self-evolution of LLMs, and we will pursue these directions in our future works.

Finally, we would like to express our deepest gratitude to the AC and all reviewers for their efforts in reviewing the manuscript and for their willingness to engage in further discussion!

Best Wishes,

Authors of manuscript #28761

---

### Decision · Program_Chairs · 2025-09-17

**Decision:**

Accept (poster)

**Comment:**

This paper proposes a heterogeneous large language model fusion framework using synthetic data only. It organizes domain knowledge into a three-level tree, formulates domain expansion and data ratio allocation as a hierarchical multi-armed bandit (HMAB) problem. The   framework also uses DynaBranches mechanism with an introspection-rebirth (IR) module that adaptively adjusts domain sampling using a sliding window binomial likelihood ratio test (SWBLRT).


The proposed framework seems very complicated and the reviewers raised concerns about the implementation details. Also, the experiment setup (the characteristics of the set of source LLMs) seem narrow. Additional information and experiment results can make it paper better.

Overall, this is a solid work but I am conservative for its real impact.